# CSGO: Content-Style Composition in Text-to-Image Generation

**Peng Xing**[1,2]    **Haofan Wang**[2]    **Yanpeng Sun**[1]    **Qixun Wang**[2]
**Xu Bai**[2,3]    **Hao Ai**[4]    **Jen-Yuan Huang**[5]    **Zechao Li**[1,†]
[1]Nanjing University of Science and Technology    [2]InstantX Team    [3]Xiaohongshu Inc
[4]Beihang University    [5]Peking University
{xingp_ng,yanpeng_sun,zechao.li}@njust.edu.cn
haofanwang.ai@gmail.com
[†] Corresponding author

## Abstract

The advancement of image style transfer has been fundamentally constrained by the absence of large-scale, high-quality datasets with explicit content-style-stylized supervision. Existing methods predominantly adopt training-free paradigms (e.g., image inversion), which limit controllability and generalization due to the lack of structured triplet data. To bridge this gap, we design a scalable and automated pipeline that constructs and purifies high-fidelity content-style-stylized image triplets. Leveraging this pipeline, we introduce IMAGStyle—the first large-scale dataset of its kind, containing 210K diverse and precisely aligned triplets for style transfer research. Empowered by IMAGStyle, we propose CSGO, a unified, end-to-end trainable framework that decouples content and style representations via independent feature injection. CSGO jointly supports image-driven style transfer, text-driven stylized generation, and text-editing-driven stylized synthesis within a single architecture. Extensive experiments show that CSGO achieves state-of-the-art controllability and fidelity, demonstrating the critical role of structured synthetic data in unlocking robust and generalizable style transfer. Source code: https://github.com/instantX-research/CSGO

## 1   Introduction

Recent advancements in diffusion models have significantly improved the field of text-to-image generation [39, 16]. Models such as SD [29] excel at creating visually appealing images based on textual prompts, playing a crucial role in personalized content creation [31, 47, 37]. Despite numerous studies on general controllability, image style transfer remains particularly challenging.

Image style transfer aims to generate a plausible target image by combining the content of one image with the style of another, ensuring that the target image maintains the original content's semantics while adopting the desired style [20, 9]. This process requires fine-grained control over content and style, involving abstract concepts like texture, color, and visual quality, making it a complex and nuanced challenge [7].

A significant challenge in style transfer is the lack of a large-scale stylized dataset, which makes it impossible to train models end-to-end and results in suboptimal style transfer quality for non-end-to-end methods. Existing methods typically rely on training-free structures, such as DDIM inversion [39] or carefully tuned feature injection layers of pre-trained IP-Adapter [48]. Methods like Plug-and-Play [40], VCT [4], and the state-of-the-art StyleID [7] employ content image inversion and sometimes style image inversion to extract and inject image features into specifically designed layers. However, inverting content and style images significantly increases inference time, and DDIM

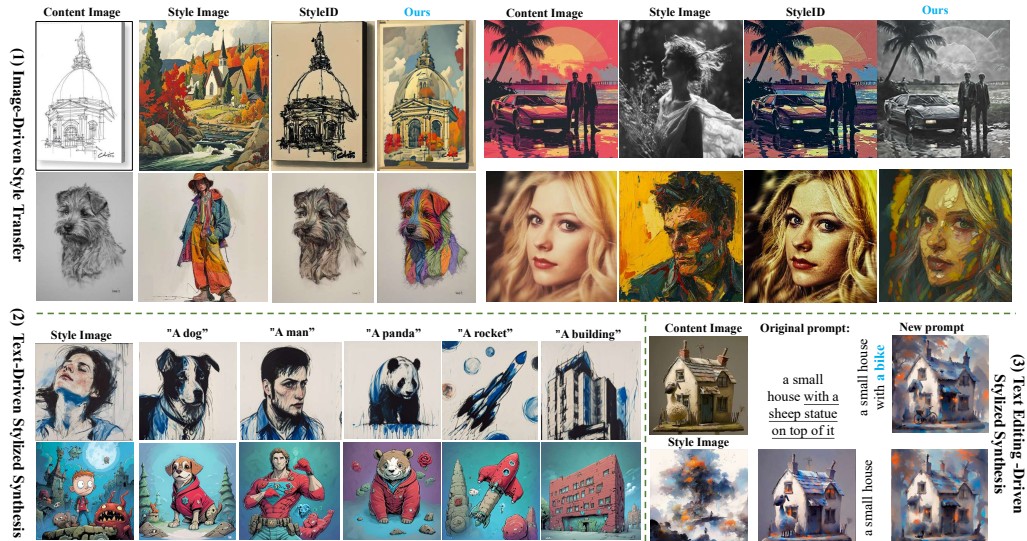

Figure 1: (1) Comparison of the style transfer results of the proposed method with the recent state-of-the-art method StyleID [7]. (2) Our CSGO achieves high-quality text-driven stylized synthesis. (3) Our CSGO achieves high-quality text editing-driven stylized synthesis.

inversion can lose critical information [22], leading to failures, as shown in Figure 1. InstantStyle [41] employs the pre-trained IP-Adapter. However, it struggles with accurate content control. Another class of methods relies on a small amount of data to train LoRA and implicitly decouple content and style LoRAs, such as ZipLoRA [35] and B-LoRA [12], which combine style and content LoRAs to achieve content retention and Style transfer. However, each image requires fine-tuning, and implicit decoupling reduces stability.

To overcome the above challenges, we start by constructing a style transfer-specific dataset and then design a simple yet effective framework to validate the beneficial effects of this large-scale dataset on style transfer. Initially, we propose a dataset construction pipeline for Content-Style-Stylized Image Triplets (CSSIT), incorporating both a data generation method and an automated cleaning process. Using this pipeline, we construct a large-scale stylized dataset, **IMAGStyle**, comprising 210K content-style-stylized image triplets. Next, we introduce an end-to-end trained style transfer framework, **CSGO**. Unlike previous implicit extractions, it explicitly uses independent content and style feature injection modules to achieve high-quality image style transformations. The framework simultaneously accepts style and content images as inputs and efficiently fuses content and style features using well-designed feature injection blocks. Benefiting from the decoupled training framework, once trained, CSGO realizes any form of arbitrary style transfer without fine-tuning at the inference stage, including sketch or nature image-driven style transfer, text-driven, text editing-driven stylized synthesis. Finally, we utilize Content Alignment Score (CAS) and CSD [38] score to evaluate the quality of style transfer, effectively measuring the degree of content and style loss post-transfer. Extensive qualitative and quantitative studies validate that our proposed method achieves advanced zero-shot style transfer.

## 2 Related Work

Style transfer has garnered significant attention and research due to its practical applications in art creation [14, 36, 42]. Early methods, both optimization-based [14] and inference-based [2, 11], are limited by speed constraints and the diversity of style transfer. The AdaIN approach [18], which separates content and style features from deep features, has become a representative method for style transfer, inspiring a series of techniques using statistical mean and variance [3, 15]. Additionally, transformer-based methods such as StyleFormer [46] and StyTR$^2$ [8] improve content bias. However, these methods primarily focus on color or stroke transfer and face limitations in arbitrary style transfer.

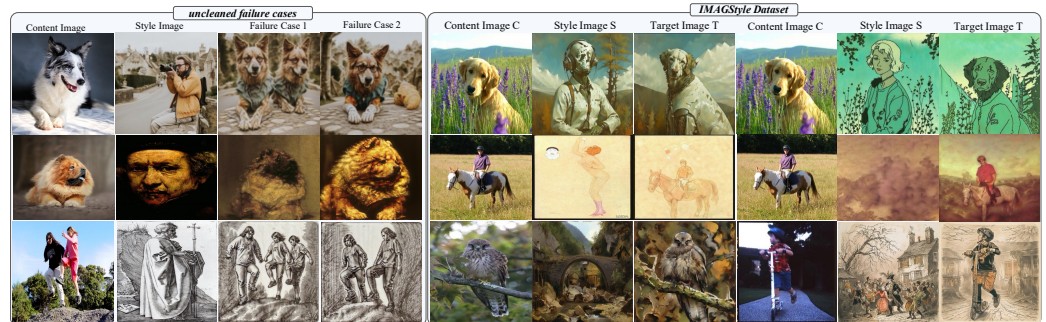

Figure 2: (a) Left: Failure cases in step (1), which fail to maintain the spatial structure of the content image. (b) Right: Samples from our IMAGStyle dataset.

Currently, inversion-based Style Transfer (InST) [50] is proposed to obtain inversion latent of style image and manipulate attention maps to edit generated Images. However, DDIM (Denoising Diffusion Implicit Models) inversion results in content loss and increased inference time [39]. Hertz *et al.* explore self-attention layers using key and value matrices for style transfer [15]. DEADiff [26] and StyleShot [13] are trained through a two-stage style control method. However, it is easy to lose detailed information within the control through sparse lines. InstantStyle [41, 42] to achieve high-quality style control through pre-trained prompt adapter [48] and carefully designed injection layers. However, these methods struggle with achieving high-precision style transfer and face limitations related to content preservation. Some fine-tuning approaches, such as LoRA [17], DB-LoRA [32], Zip-LoRA [35], and B-LoRA [12], enable higher-quality style-controlled generation but require fine-tuning for different styles and face challenges in achieving style transfer. Our proposed method introduces a novel style transfer dataset and develops the CSGO framework, achieving high-quality arbitrary image style transfer without the need for fine-tuning.

## 3 Data Pipeline

In this section, we first introduce the proposed pipeline for constructing content-style-stylized image triplets. Then, we describe the constructed IMAGStyle dataset in detail.

### 3.1 Pipeline for Constructing Image Triplets

The lack of a large-scale open-source dataset of content-style-stylized image pairs (CSSIT) in the community seriously hinders the research on style transfer. In this work, we propose a data construction pipeline that automatically constructs and cleans to obtain high-quality content-style-stylized image triplets, given only arbitrary content images and style images. The pipeline contains two steps: (1) stylized image generation and (2) stylized image cleaning.

**Stylized image generation.** Given an arbitrary content image $C$ and an arbitrary style image $S$, the goal is to generate a stylized image $T$ that preserves the content of $C$ while adopting the style of $S$. We are inspired by B-LoRA [12], which finds that content LoRA and style LoRA can be implicitly separated by SD-trained LoRA, preserving the original image's content and style information, respectively. Therefore, we first train a large number of LoRAs with lots of content and style imges. To ensure that the content of the generated image $T$ is aligned to $C$ as much as possible, the loRA for $C$ is trained using only one content image $C$. Then, Each trained loRA is decomposed into a content LoRA and a style LoRA through implicit separate mentioned by work [12]. Finally, the content LoRA of image $C$ is combined with

---

**Algorithm 1** Pipeline of Constructing CSSIT

**Input:** content images $Set_{content}$, style images $Set_{style}$
**Output:** Content-style-stylized image triplets $Set$
1: **for** each $C \in Set_{content}$ **do**
2:     $C_{LoRA} \leftarrow$ Train LoRA for $C$
3:     $C_{LoRA}^{content} \leftarrow$ Separate content LoRA in $C_{LoRA}$
4:     **for** each $S \in Set_{style}$ **do**
5:         $S_{LoRA} \leftarrow$ Train LoRA for $S$
6:         $S_{LoRA}^{style} \leftarrow$ Separate style LoRA in $S_{LoRA}$
7:         $CS_{LoRA} \leftarrow$ Combine $C_{LoRA}^{content}$ and $S_{LoRA}^{style}$
8:         $T = \{T_1, T_2, ..., T_n\} \leftarrow$ Generate $n$ images by $CS_{LoRA}$
9:         $CAS_1, CAS_2, ..., CAS_n \leftarrow$ Compute CAS for each generated image based on Equ.( 1)
10:         $i \leftarrow$ Obtain the index of the minimum value of all CAS
11:         $Set$.append([$C, S, T_i$])
12:     **end for**
13: **end for**
14: **return** $Set$

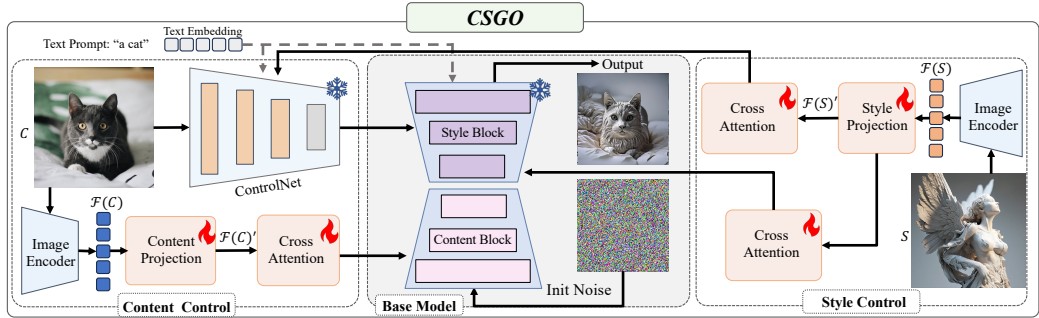

Figure 3: Overview of the proposed end-to-end style transfer framework CSGO.

the style LoRA of $S$ to generate the target images $T = \{T_1, T_2, ..., T_n\}$ using the base model. However, the implicit separate approach is unstable, resulting in the content and style LoRA not reliably retaining content or style information. This manifests itself in the form of the generated image $T_i$, which does not always agree with the content of $C$, as shown in Figure 2 (left). Therefore, it is necessary to filter $T$, sampling the most reasonable $T_i$ as the target image.

**Stylized image cleaning.** Slow methods of cleaning data with human involvement are unacceptable for the construction of large-scale stylized data triplets. To this end, we develop an automatic cleaning method to obtain the ideal and high-quality stylized image $T$ efficiently. First, we propose a content alignment score (CAS) that effectively measures the content alignment of the generated image with the content image. It is defined as the feature distance between the content semantic features (without style information) of the generated image and the original content image. It is represented as follows:

$$CAS_i = \|Ada(\phi(C)) - Ada(\phi(T_i))\|^2 , \tag{1}$$

where $CAS_i$ denotes the content alignment score of generated image $T_i$, $\phi(\cdot)$ denotes image encoder. We compare the mainstream feature extractors and the closest to human filtering results is DINO-V2 [21]. $Ada(F)$ represents a function of feature $F$ to remove style information. We follow AdaIN [18] to express style information by mean and variance, $Ada(F) = \frac{F - \mu(F)}{\rho(F)}$, where $\mu(F)$ and $\rho(F)$ represent the mean and variance of feature $F$. Obviously, a smaller CAS indicates that the generated image is closer to the content of the original image. In Algorithm 1, we provide a pseudo-code of our pipeline.

## 3.2 IMAGStyle Dataset Details

**Content Images.** To ensure that the content images have clear semantic information and facilitate separating after training, we employ the saliency detection datasets, MSRA10K [5, 6] and MSRA-B [19], as the content images. In addition, for sketch stylized, we sample 1000 sketch images from ImageNet-Sketch [43] as content images. We use BLIP [21] to generate a caption for each content image. A total of 11,000 content images are trained and used as content LoRA.

**Style Images.** To ensure the richness of the style diversity, we sample 5000 images of different painting styles (history painting, portrait, genre painting, landscape, and still life) from the Wikiart dataset [33]. In addition, we generated 5000 images using Midjourney covering diverse styles, including Classical, Modern, Romantic, Realistic, Surreal, Abstract, Futuristic, Bright, Dark, etc. A total of 10,000 style images are used to train style LoRA.

**Dataset.** Based on the pipeline described in Section 3.1, as shown in Figure 2 (right), we construct a style transfer dataset, **IMAGStyle**, which contains 210K content-style-stylized image triplets as training dataset. Furthermore, we collect 248 content images from the web containing images of real scenes, sketched scenes, faces, and style scenes, as well as 206 style images of different scenes as testing dataset. For testing, each content image is transferred to 206 styles. This dataset will be used for community research on style transfer and stylized synthesis.

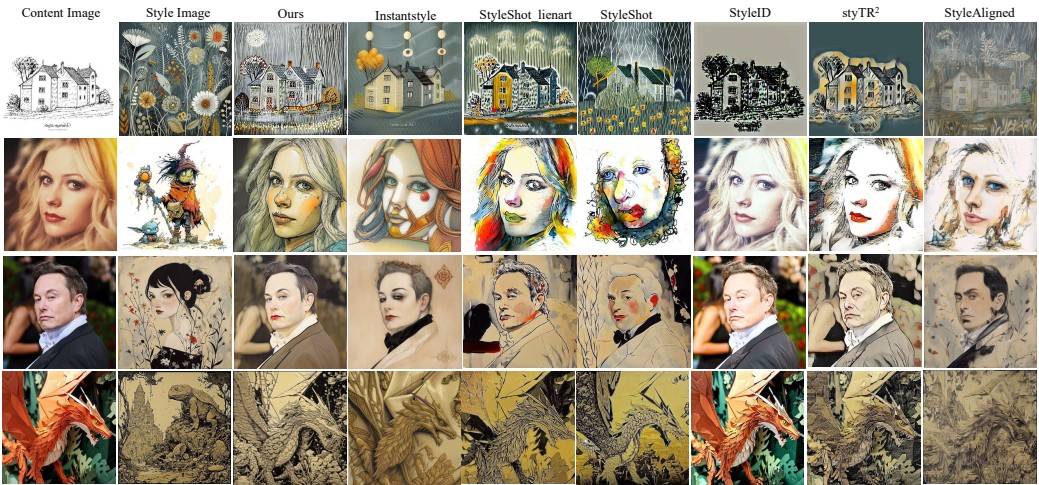

Figure 4: Comparison of image-driven style transfer results. Zoomed in for the best viewing.

# 4 Approach

## 4.1 CSGO framework

The proposed style transfer model, CSGO, shown in Figure 3, aims to achieve arbitrary stylization of any image without fine-tuning, including sketch and natural image-driven style transfer, text-driven stylized synthesis, and text editing-driven stylized synthesis. Benefiting from the proposed IMAGStyle dataset, the proposed CSGO supports an end-to-end style transfer training paradigm. To ensure effective style transfer and accurate content preservation, we carefully design the content and style control modules. In addition, to reduce the risk that the content image leaks style information or the style image leaks content, the content control and style control modules are explicitly decoupled, and the corresponding features are extracted independently. To be more specific, we categorize our CSGO into two main components and describe them in detail.

Table 1: Comparison with recent state-of-the-art methods on the test dataset.

|  | StyTR$^2$ [8] | Style-Aligned [15] | StyleID [7] | InstantStyle [41] | StyleShot [13] | StyleShot-lineart [13] | CSGO |
|---|---|---|---|---|---|---|---|
| CSD ($\uparrow$) | 0.2695 | 0.4274 | 0.0992 | 0.3175 | 0.4522 | 0.3903 | 0.5146 |
| CAS ($\downarrow$) | 0.9699 | 1.3930 | 0.4873 | 1.3147 | 1.5105 | 1.0750 | 0.8386 |
| AS ($\uparrow$) | 4.0387 | 3.7463 | 4.7643 | 5.4824 | 5.6728 | 5.2542 | 5.5467 |

**Content Control.** The purpose of content control is to ensure that the stylized image retains the semantics, layout, and other features of the content image. To this end, we carefully designed two ways of content control. First, we implement content control through pre-trained ControlNet [49], whose input is the content image and the corresponding caption. We leverage the capabilities of the specific content-controllable model(Tile ControlNet) to reduce the data requirements and computational

Table 2: User Preference Score.

| VS | Win | Tie | Loss |
|---|---|---|---|
| StyleShot[13] | 58.5 | 21.4 | 20.1 |
| InstantStyle[41] | 64.2 | 20.6 | 15.2 |
| StyleAligned [15] | 77.0 | 12.3 | 10.7 |

costs of training content retention from scratch Following the ControlNet, the output of ControlNet is directly injected into the up-sampling blocks of the base model (pre-trained UNet in SD) to obtain fusion output $D'_i = D_i + \delta_c \times C_i$, $D_i$ denotes the output of $i$-th block in the base model, $C_i$ denotes the output of $i$-th block in ControlNet, $\delta_c$ represents the fusion weight.

In addition, to achieve content control in the down-sampling blocks of the base model, we utilize an additional learnable cross-attention layer to inject content features into down blocks. Specifically, we use pre-trained CLIP image encoder [27] and a learnable projection layer to extract the semantic feature $\mathcal{F}(C)'$ of the content image. Then, we utilize an additional cross-attention layer to inject the extracted content features into the down-sampling blocks, *i.e.*, $D'_C = D + \lambda_c \times D_C$, $D$ denotes the output of in the base model, $D_C$ denotes the output of content IP-Adapter, $\lambda_c$ represents the fusion weight [48]. These two content control strategies ensure small content loss during the style transfer.

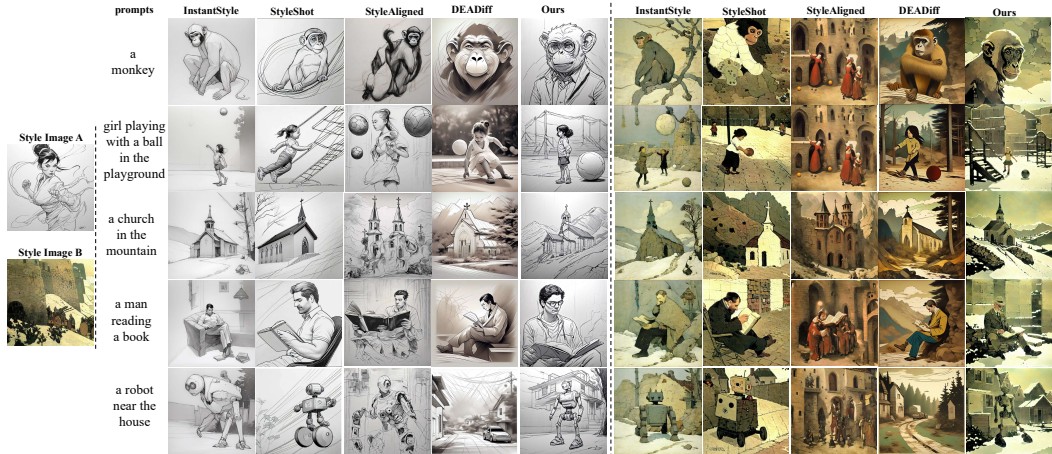

Figure 5: Comparison of generation results for text-driven stylized synthesis with recent methods.

**Style Control.** To ensure that the proposed CSGO has strong style control capability, we also design two simple yet effective style control methods. Generally, we feed the style images into a pre-trained image encoder to extract embedding $\mathcal{F}(S) \in \mathbb{R}^{o \times d}$ and map them to the style embedding $\mathcal{F}(S)' \in \mathbb{R}^{t \times d}$ through the Perceiver Resampler structure [1]. Here, $o$ and $t$ represent the token number of original and style embeddings, $d$ denotes the dimension of $\mathcal{F}(S)$. Then, we utilize an new cross-attention layer to inject the style embedding into the up-sampling blocks of the base model.

Furthermore, we note that relying only on the injection of the up-sampling blocks of the base model weakens the style control since ControlNet injections in the content control may leak style information of the content image $C$. For this reason, we propose to use an independent cross attention module to simultaneously inject style features into, and the fusion weight is $\lambda_s$, as shown in Figure 3. The insight of this is to pre-adjust the style of the content image using style features making the output of the Controlnet model retain the content while containing the desired style features.

In summary, the proposed CSGO framework explicitly learns separate feature processing modules that inject style and content features into different locations of the base model, respectively. Despite its simplicity, CSGO achieves state-of-the-art style transfer results.

### 4.2 Model Training and Inference.

**Training.** Based on the proposed dataset, IMAGStyle, our CSGO is the first implementation of end-to-end style transfer training. Given a content image $C$, a caption $P$ of the content image, a style image $S$, and a target image $T$, we train a style transfer network based on a pre-trained diffusion model. Our training objective is to model the relationship between the styled image $T$ and Gaussian noise under content and style image conditions, which is represented as follows:

$$\mathcal{L} = \mathbb{E}_{z_0,t,P,C,S,\epsilon \sim \mathcal{N}(0,1)} \left[ \|\epsilon - \epsilon_\theta \left( z_t, t, C, S, P \right)\|^2 \right], \tag{2}$$

where $\varepsilon$ denotes the random sampled Gaussian noise, $\varepsilon_\theta$ denotes the trainable parameters of CSGO, $t$ represents the timestep. Note that the latent latent $z_t$ is constructed with a style image $T$ during training, $z_t = \sqrt{\bar{\alpha}_t}\psi(T) + \sqrt{1 - \bar{\alpha}_t}\varepsilon$, where $\psi(\cdot)$ mapping the original input to the latent space function, $\bar{\alpha}_t$ is consistent with diffusion models [39, 16]. We randomly drop content image and style image conditions in the training phase to enable classifier-free guidance in the inference stage.

## 5 Experiments

### 5.1 Experimental Setup

**Setup.** For the IMAGstyle dataset, during the training phase, we suggest using 'a [vcp]' as a prompt for content images and 'a [stp]' as a prompt for style images. The rank is set to 64 and each B-loRA is trained with 1000 steps. During the generation phase, we suggest using 'a [vcp] in [stv] style' as the prompt. For the CSGO framework, we employ

*stabilityai/stable-diffusion-xl-base-1.0* as the base model, pre-trained *ViT-H* as image encoder, and *TTPlanet/TTPLanet_SDXL_Controlnet_Tile_Realistic* as ControlNet. we uniformly set the images to $512 \times 512$ resolution. The drop rate of text, content image, and style image is 0.15. The learning rate is 1e-4. During training stage, $\lambda_c = \lambda_s = \delta_c = 1.0$. During inference stage, we suggest $\lambda_c = \lambda_s = 1.0$ and $\delta_c = 0.5$. Our experiments are conducted on 8 NVIDIA H800 GPUs (80GB) with a batch size of 20 per GPU and trained 80000 steps.

**Datasets and Evaluation.** We use the proposed IMAGStyle as a training dataset and use its testing dataset as an evaluation dataset. It is worth noting that the style transfer task, unlike the rest of the style control tasks, requires a trade-off between content retention and style quality at the same time. We use the CSD score [38] as an evaluation metric to evaluate the style similarity and Aesthetic Score as a quality evaluation metric. Meanwhile, we employ the content alignment score (CAS) as an evaluation metric to evaluate the content similarity.

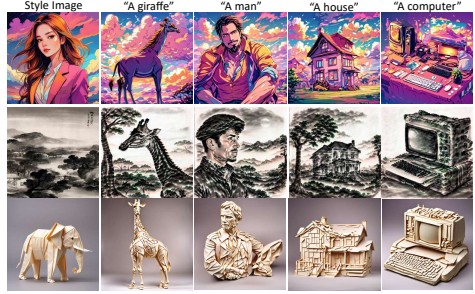

Figure 6: Generated results of the proposed CSGO in text-driven stylized synthesis.

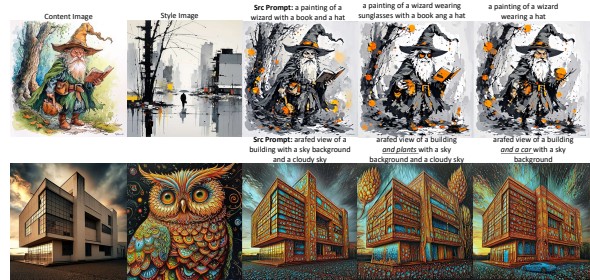

Figure 7: The generated results of the proposed CSGO in text editing-driven stylized synthesis.

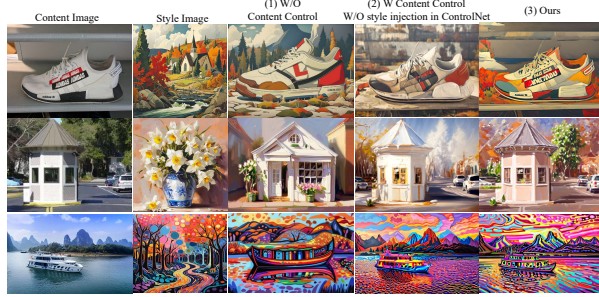

Figure 8: Ablation studies of content control and style control.

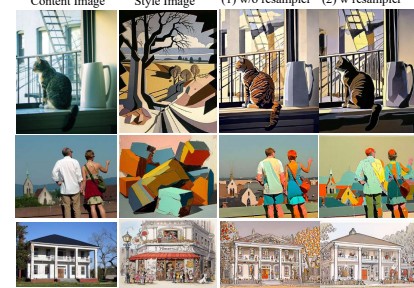

Figure 9: Ablation studies of style image projection.

**Baselines.** We compare recent advanced inversion-based StyleID [7], StyleAligned [15] methods, and StyTR$^2$ [8] based on the Transformer structure. In addition, we compare Instantstyle [41] and StyleShot (and their fine-grained control method StyleShot-lineart) [13] that introduce ControlNet and IPAdapter structures as baselines. For text-driven style control task, we also introduce DEADiff [26] as a baseline.

## 5.2 Experimental Results

**Image-Driven Style Transfer.** In Table 4, we demonstrate the CSD scores and CAS of the proposed method with recent advanced methods for the image-driven style transfer task. In terms of style control, our CSGO achieves the highest CSD score, demonstrating that CSGO achieves state-of-the-art style control. Due to the decoupled style injection approach, the proposed CSGO effectively extracts style features and fuses them with high-quality content features. As illustrated in Figure 4, Our CSGO precisely transfers styles while maintaining the semantics of the content in natural, sketch, face, and art scenes. More results for style control can be found in the supplementary material.

In terms of content retention, it can be observed that StyleID [7] and StyleAligned [15], which are based on inversion, maintain the original content too strongly in sketch style transfer scenarios (CAS is very low). However. they are unable to inject style information since CSD score is low. InstantStyle [41] and StyleShot [13] (including Lineart), which use lines to control the content, are

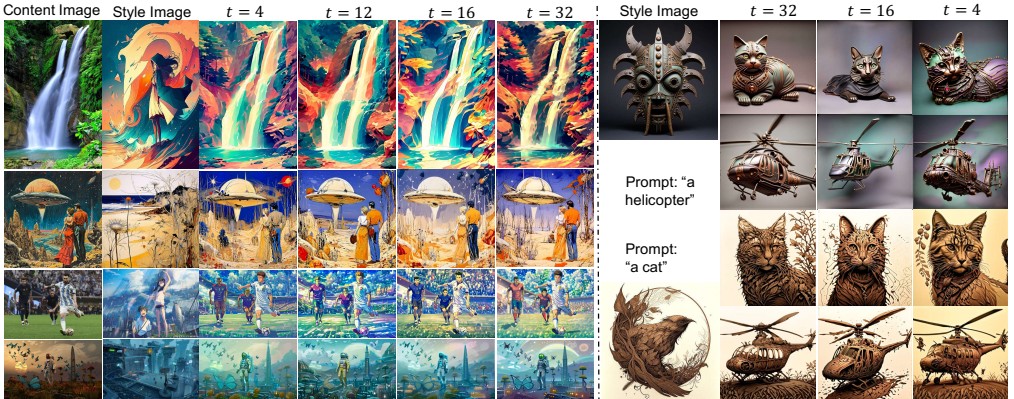

Figure 10: Ablation studies of style token number $t$. Left: Image style transfer results. Right: Text-driven stylized synthesis results.

affected by the level of detail of the lines and have different degrees of loss of content (such as face scenes). The proposed CSGO directly utilizes all the information of the content image, and content preservation is optimal. The quantitative results in Table 4 also show that the proposed CSGO maintains high-quality content retention with precise style transfer. It is worth noting that it is possible to implement a content-style cycle transfer in CSGO (as shown in Figure 12; see the Appendix for more results).

**Text-Driven Stylized Synthesis.** The proposed method enables text-driven style control, *i.e.*, given a text prompt and a style image, generating images with similar styles. Figure 5 shows the comparison of the generation results of the proposed CSGO with the state-of-the-art methods. In a simple scene, it is intuitive to observe that our CSGO obeys textual prompts more. The reason for this is that thanks to the explicit decoupling of content and style features, style images only inject style information without exposing content. In addition, in complex scenes, thanks to the well-designed style feature injection block, CSGO enables optimal style control while converting the meaning of text. As illustrated in Figure 6, we demonstrated more results.

**Text editing-Driven Stylized Synthesis.** The proposed CSGO supports text editing-driven style control. As shown in Figure 7, in the style transfer, we maintain the semantics and layout of the original content images while allowing simple editing of the textual prompts. The above excellent results demonstrate that the proposed CSGO is a powerful framework for style control.

**User Preference.** We randomly selected 100 sets of results from the test set. Of these, 20 were portraits and 20 were sketches, and the rest were randomized. We then conducted a user study experiment comparing CSGO with Styleshot-lineart, instantStyle, and Stylealigned. Each group contained four generated results, and the user chose the best result. The results are shown in Table **??**, where it can be noticed that CSGO gets the highest preference score.

## 5.3 Ablation Studies.

**Content control and style control.** We discuss the impact of the two feature injection methods, as shown in Figure 8 and Table 3. If content images are injected into the base model only through an additional cross attention layer, only semantic information is guaranteed, while the full content information is not preserved (Figure 8(1)). After introducing the ControlNet injection, the quality of content retention improved, as shown in Figure 8. However, if the style features are injected into base UNet only without ControlNet injection, this weakens the style of the generated images, which can be observed in the comparison of Figure 8(2) and (3).

Therefore, the proposed CSGO pre-injects style features in the ControlNet branch to further fuse the style features to enhance the transfer effect.

**Style image projection layer.** The style image projection layer can effectively extract style features from the original embedding. We explore the normal linear layer and the Resampler structure, and the experimental results are shown in Figure 9. Using the Resampler structure captures more detailed style features while avoiding content leakage.

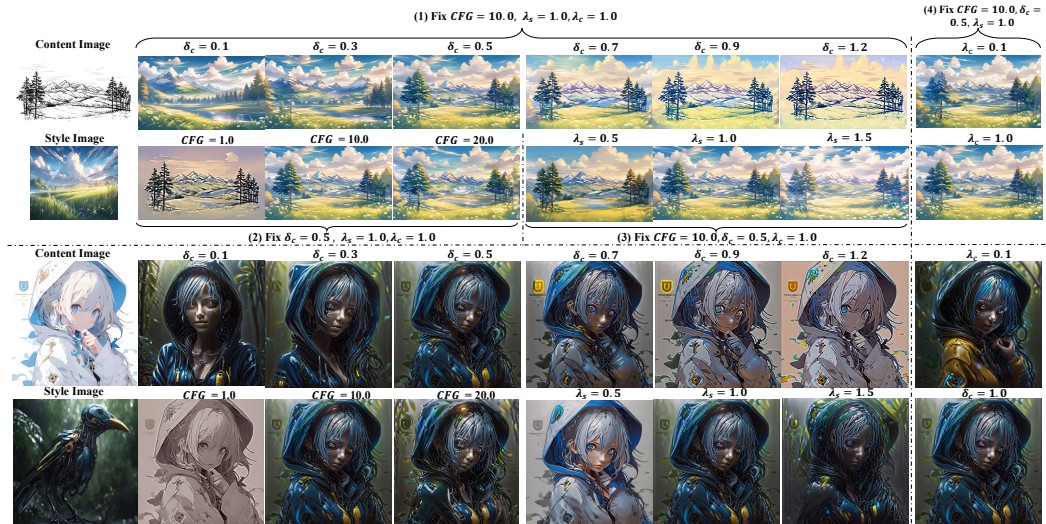

Figure 11: Ablation studies of content scale $\delta_c$, CFG, content scale $\lambda_c$, and style scale $\lambda_s$.

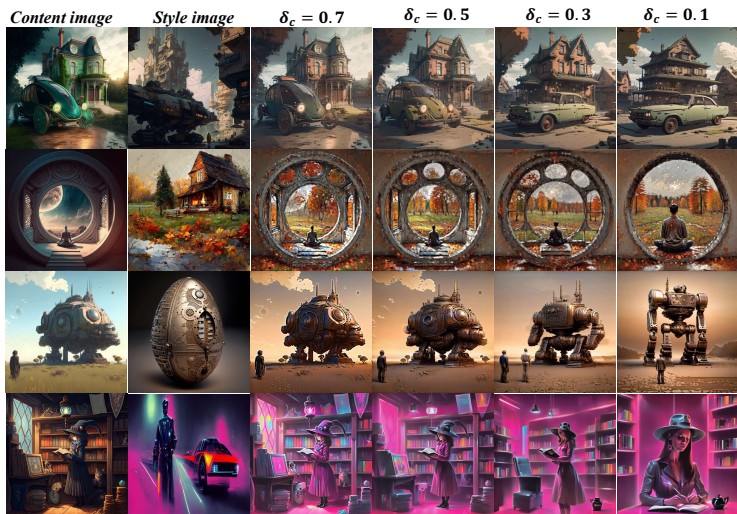

Figure 12: Effects of content scale in style transfer task.

**Token number.** We explore the effect of the number of token $t$ in the style projection layer on the results of style transfer and text-driven style synthesis. The experimental results are shown in Figure 9, where the style control becomes progressively better as $t$ increases. This is in line with our expectation that $t$ influences the quality of feature extraction. A larger $t$ means that the projection layer can extract richer style features.

Table 3: Ablation results on the test dataset.

|  | W/O Content Control | W/O Style Injection | W/O Resampler | CSGO |
|---|---|---|---|---|
| CSD ($\uparrow$) | 0.5381 | 0.4873 | 0.4926 | **0.5146** |
| CAS ($\downarrow$) | 1.7723 | 0.8372 | 0.8279 | **0.8386** |

**The impact of content scale $\delta_c$.** As shown in Figure 11, when $\delta_c$ is small, the content feature injection is weak, and CSGO obeys the textual prompts and style more. As $\delta_c$ increases, the quality of content retention becomes superior. However, we notice that when $\delta_c$ is large (e.g., 0.9 and 1.2), the style information is severely weakened.

**The impact of CFG scale.** Classifier-free guidance enhances the capabilities of the text-to-image model. The proposed CSGO is similarly affected by the strength of CFG scale. As shown in Figure 11, the introduction of CFG enhances the style transfer effect.

**The impact of style scale $\lambda_s$ and content scale $\lambda_c$.** The style scale affects the degree of style

injection. Figure 11 shows that if the style scale is less than 1.0, the style of the generated image is severely weakened. We suggest that the style scale should be between 1.0 and 1.5. Content control in the down-sampling blocks utilizes the semantic information of the content image to reinforce the accurate retention of content. Figure 11 shows that $\lambda_c$ is most effective when it is near 1.0. In style transfer, the retention of content and style varies from person to person. We can set the hyperparameter content scale $\delta_c$ so that the generated result meets the expectation. As shown in Figure 12, different levels of detailed information can be retained by different content scales to meet different design requirements.

## 6   Conclusion

We first propose a pipeline for the construction of content-style-stylized image triplets. Based on this pipeline, we construct the first large-scale style transfer dataset, IMAGStyle, which contains 210K image triplets and covers a wide range of style scenarios. To validate the impact of IMAGStyle on style transfer, we propose CSGO, a simple but highly effective end-to-end training style transfer framework, and we verify that the proposed CSGO can simultaneously perform image style transfer, text-driven style synthesis, and text editing-driven style synthesis tasks in a unified framework. Extensive experiments validate the beneficial effects of IMAGStyle and CSGO for style transfer. We hope that our work will inspire the research community to further explore stylized research.

## 7   Acknowledgments

This work was supported by National Natural Science Foundation of China (Grant No. 62425603) and Basic Research Program of Jiangsu Province (Grant No. BK20240011).

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

# Appendix: CSGO: Content-Style Composition in Text-to-Image Generation

## A  Details.

### A.1  Preliminaries

**Text-to-Image Model.** In recent years, diffusion models have garnered significant attention in the text-to-image generation community due to their powerful generative capabilities demonstrated by early works [10, 28]. Owing to large-scale training [34], improved architectures [27, 24], and latent space diffusion mechanisms, models like Stable Diffusion have achieved notable success in text-to-image generation. [28]. The focus on controllability in text-to-image models has grown in response to practical demands. Popular models such as ControlNet [49], T2Iadapter [23], and IP-Adapter [48] introduce additional image conditions to enhance controllability. These models use sophisticated feature extraction methods and integrate these features into well-designed modules to achieve layout control. In this paper, we present a style transfer framework, CSGO, based on an image-conditional generation model that can perform zero-shot style transfer.

**Stable Diffusion.** The backbone of the stable diffusion model [29] uses CLIP [27] as a text encoder and the UNet structure [30] as a latent denoising network. We refer to the pre-trained UNet as the base model. In general, the U-Net contains multiple down-sampling blocks, a middle block, and multiple up-sampling blocks [25]. Studies on the controllability of diffusion models usually inject control features into the base model.

**ControlNet.** ControlNet models have been developed for image conditions such as depth map, canny, sketch, which effectively enhance the controllability of text-to-image models. ControlNet [49] takes images as the control condition, trains zero-convolution layers and replicates encoder (down-sampling blocks and middle block) of the base model, and injects the resulting outputs of each block into the up-sampling blocks and middle block correspondingly. Controlnet output features are directly weighted with base model features.

**IP-Adapter.** IP-Adapter [48] implements image prompt features injected into the text-to-image model by decoupling the cross attention module. In general, image prompts are first obtained as image embeddings by a pre-trained encoder, and then mapped to the Key matrix and Value matrix of the attention. Then, they interact with the Query matrix of the base model's attention layer and weight the outputs with the original outputs. IP-Adapter's simplicity and effectiveness in injecting image conditions has received wide attention from the community.

### A.2  Datasets.

**Content Images.** We employ the saliency detection datasets, MSRA10K [5, 6] and MSRA-B [19], as the content images. In addition, we sample 1000 sketch images from ImageNet-Sketch [43] as content images to sketch-stylized. The category distribution of content images is shown in Figure 13.

In addition, we show 10 sets of CAS filtering examples in Figures 17, 18, 19, 20, 21. We show the original content image, the style image, and the best result obtained by CAS filtering (target image), and a large number of original results generated by B-LoRA. These cases show that CAS can clean illogical generated graphs for pose, size, and so on. However, we emphasize that since B-LoRA is actually more stable for the generation of styles, it is up to us to filter the images with CSD. In our experiments, it is possible to filter using only CAS without CSD.

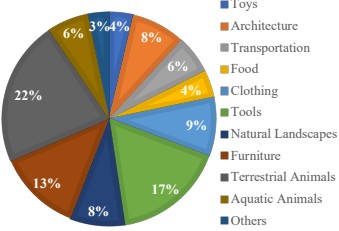

Figure 13: Distribution of content images.

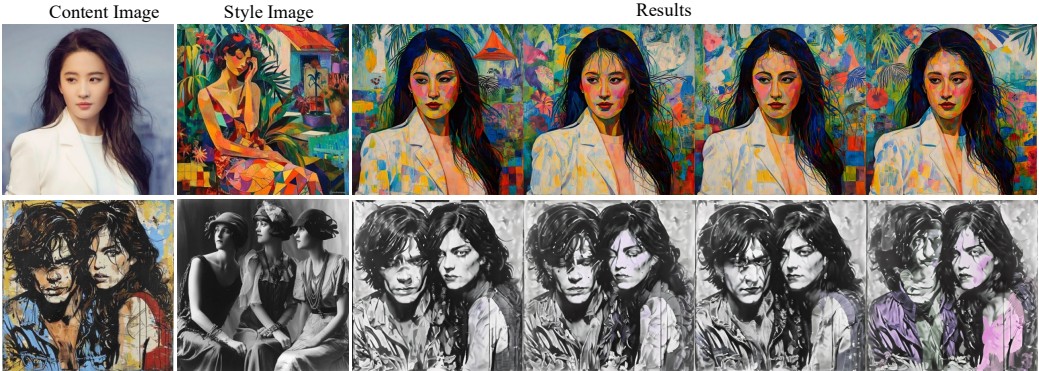

Figure 14: Failure cases.

**Setting.** In order to construct IMAGStyle, we trained a total of 11,000 content LoRAs and 10,000 style LoRAs.Theoretically, the combination of these LoRAs can generate $11,000 \times 10,000 = 110M$ images. However, image generation is very time and resource consuming. Secondly, not any combination of content LoRAs and style LoRAs can get the desired results. Therefore, we generated only 210k data triples in limited time. In the future, we expect to improve the quality of IMAGStyle further.

**Inference.** During the inference phase, we employ classifier-free guidance. The output of timestep $t$ is indicated as follows:

$$\hat{\epsilon}_\theta(z_t, t, C, S, P) = w\epsilon_\theta(z_t, t, C, S, P) + (1 - w)\epsilon_\theta(z_t, t), \tag{3}$$

where $w$ represents the classifier-free guidance factor (CFG).

## B  More Results.

### B.1  Failure cases.

As shown in Figure14, first, for real portrait stylization, as shown in the first row, there is a potential loss of facial identity. Portrait images can be difficult to collect due to the privacy issues involved, leading to some limitations in CSGO's style migration for real portraits. Second, despite incorporating styles into the ControlNet and base model, CSGO may still leak information, such as the original image's color.

In the future, we aim to enhance the CSGO framework in several ways. First, we plan to use CSGO in conjunction with LoRA to improve the portrait segment of the IMAGStyle dataset and enhance portrait stylization capabilities. Second, we will redesign and train the content extractor and style encoder to minimize content leakage. Third, we intend to refine the current IPA fusion method and explore more effective approaches to style and content fusion. However, we acknowledge that these improvements may not be achievable in the short term.

### B.2  The proposed IMAGStyle for traditional style transfer method

To comprehensively evaluate the performance of IMAGStyle, we conducted two experiments and the results in Table 4. First, we retrained StyTr$^2$ [8] using only IMAGStyle. Second, we fine-tuned StyTr$^2$ using IMAGStyle, leveraging the released model weights pre-trained for 160,000 steps. StyTr$^2$ employs a non-trivial training approach, wherein the model is implicitly constrained to produce results with content closely aligned to the content image and style closely aligned to the style image. The primary advantage of IMAGStyle lies in its <content, style, target> triplet structure. To further enhance the performance of StyTr$^2$, we introduced explicit pixel-level constraints by incorporating MSE loss. This addition enforces the generated results to be closer to the target map within the triplet, thereby improving style transfer fidelity.

Content image

Generate results of B-LoRA

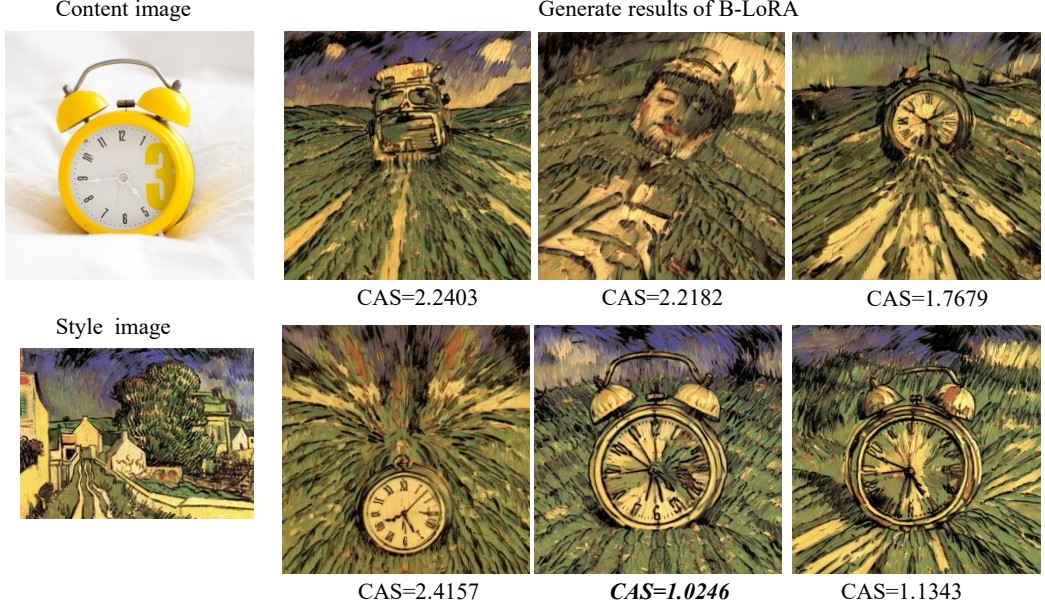

Style image

CAS=2.2403      CAS=2.2182      CAS=1.7679

CAS=2.4157      *CAS=1.0246*      CAS=1.1343

Figure 15: Example of generating a result that is filtered by CAS.

Content Image    Style Image        CSGO        B-LoRA

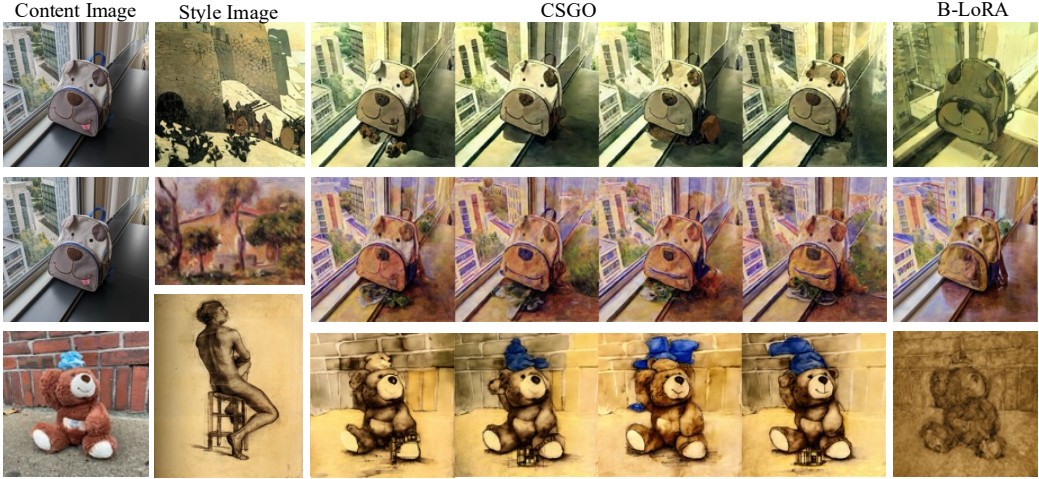

Figure 16: Comparison with LoRA.

Table 4: Comparison of style transfer metrics across different methods

| Metric | StyTr$^2$ | StyTr$^2$ with IMAGStyle | fine-tune StyTr$^2$ with IMAGStyle |
|---|---|---|---|
| CSD | 0.2695 | 0.3430 | 0.3597 |
| CAS | 0.9699 | 0.9332 | 0.9280 |
| Aesthetics Score | 4.1387 | 4.5146 | 4.6975 |

## B.3 More Results.

We show the results of additional supplementary experiments.

- **Comparison with LoRA.** LoRA-based style transfer schemes need to be fine-tuned for different style images. We compared this with the current SOTA style transfer LORA scheme B-LoRA. We trained 3 sets of B-LoRA and the results are shown in Figure 16. It can be seen that CSGO, which does not require re-fine-tuning, outperforms the style transfer results of B-LoRA.
- **Ablation studies on style token.** As expected, the number of style tokens influences the quality of style features. As shown in Figure 23, as the style token increases, the style transfer quality is better.
- **Line control and original image control.** We show in Figure 24 the results of controlling the content using lines and the original image, respectively. It can be clearly noticed that using lines loses detail information.
- **Sketch-driven style transfer results.** Figures 25, 26, and 27 show sketch-driven style transfer results.
- **Image-driven style transfer results.** Figures 28, 29, 30, and 31 shows image-driven style transfer results.
- **Content-style cycle transfer results.** Figure 32 shows content-style cycle transfer results.
- **Face image style transfer result.** Figure 33 shows face image style transfer results.
- **Text-driven stylized synthesis results.** Figure 34 and Figure 35 show text-driven stylized synthesis results.
- **Text editing-driven stylized synthesis results.** Figure 36 shows text editing-driven stylized synthesis results.

## B.4 Difference with IP-Adapter, StyleAdapter, InstantID, InstantStyle.

As shown in 5, we show the differences between CSGO and the above methods in the table below. In particular, IP-Adapter and InstantID are different from the proposed tasks applicable to CSGO. Compared to StyleAdapter and InstantStyle, CSGO support more diverse style control tasks, more detailed content and style control capabilities, and high-quality ternary style datasets.

Table 5: Method comparison summary

| Methods | IP-Adapter [48] | StyleAdapter [45] | InstantID [44] | InstantStyle [41] | CSGO |
|---|---|---|---|---|---|
| **Task** | Content consistency maintenance | Text-driven synthesis | ID consistency | Text-driven synthesis | Image-driven transfer, Text-driven synthesis, Text-editor tasks |
| **Training Data** | Reconstruction method, No pair data | Reconstruction method, No pair data | Reconstruction method, No pair data | Reconstruction method, No pair data | IMAGstyle (triplet) |
| **Structural Properties** | IP-Adapter injects features to all blocks | PCA features injected via IP-Adapter | Face features through IPA + identity net | IPAdapter weights in up_blocks.0.attentions.1 | Dual-branch content/style control via IP-Adapter and ControlNet with layer-specific feature injection. |

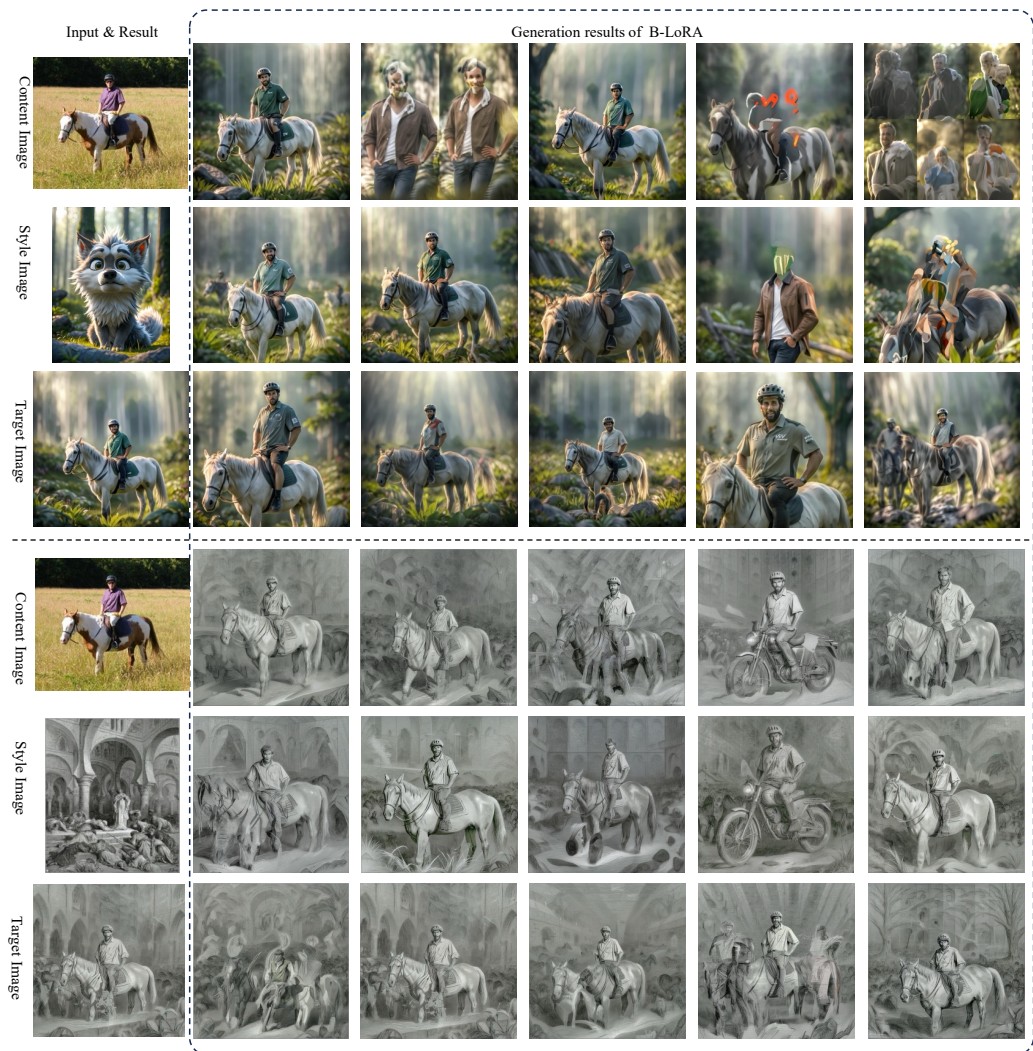

Figure 17: Example of data cleansing using CAS.

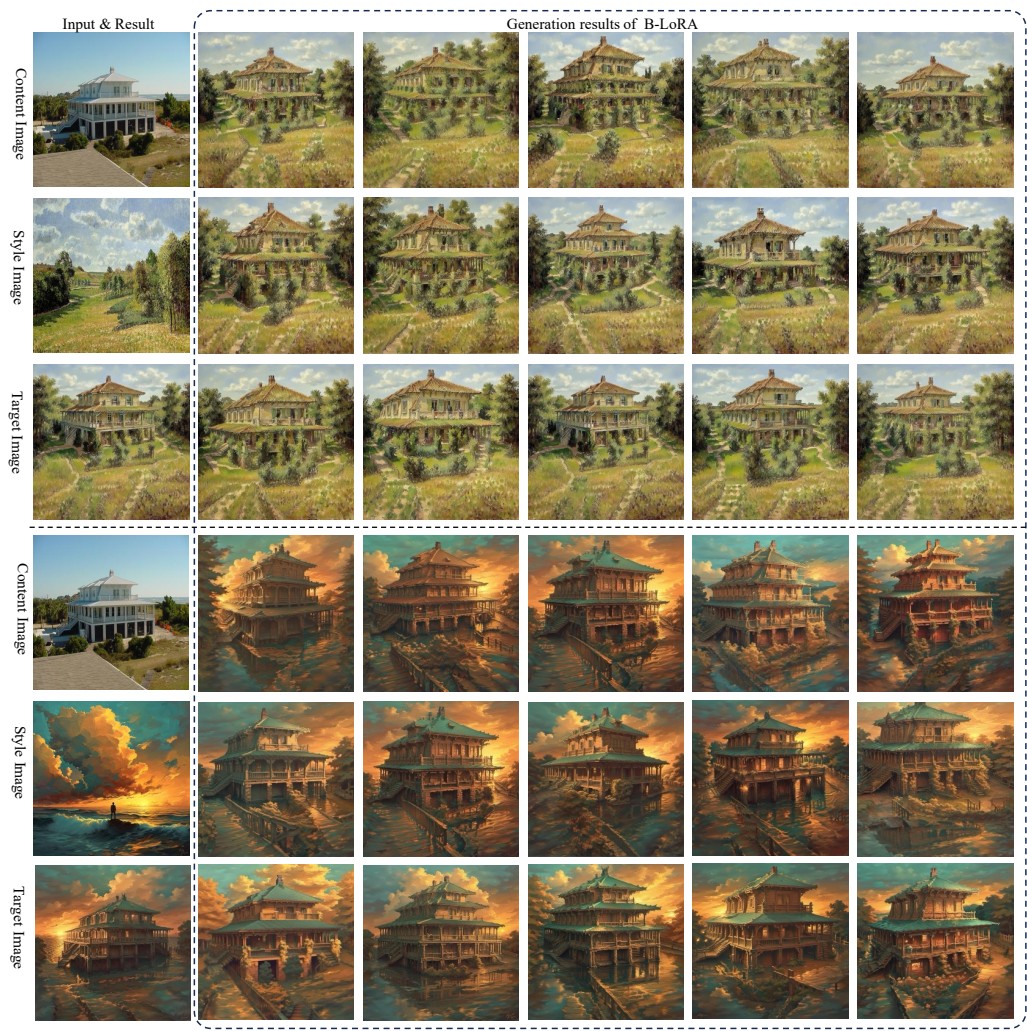

Figure 18: Example of data cleansing using CAS.

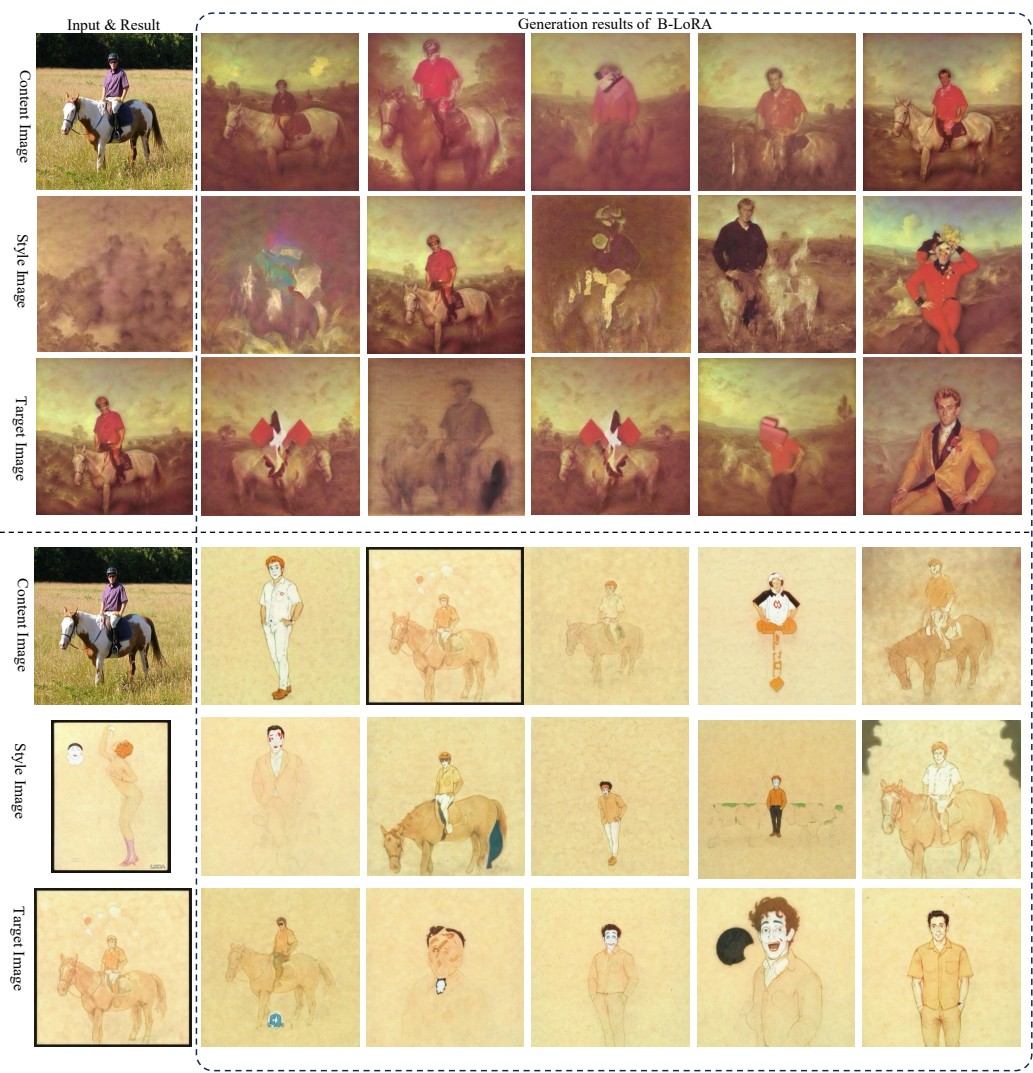

Figure 19: Example of data cleansing using CAS.

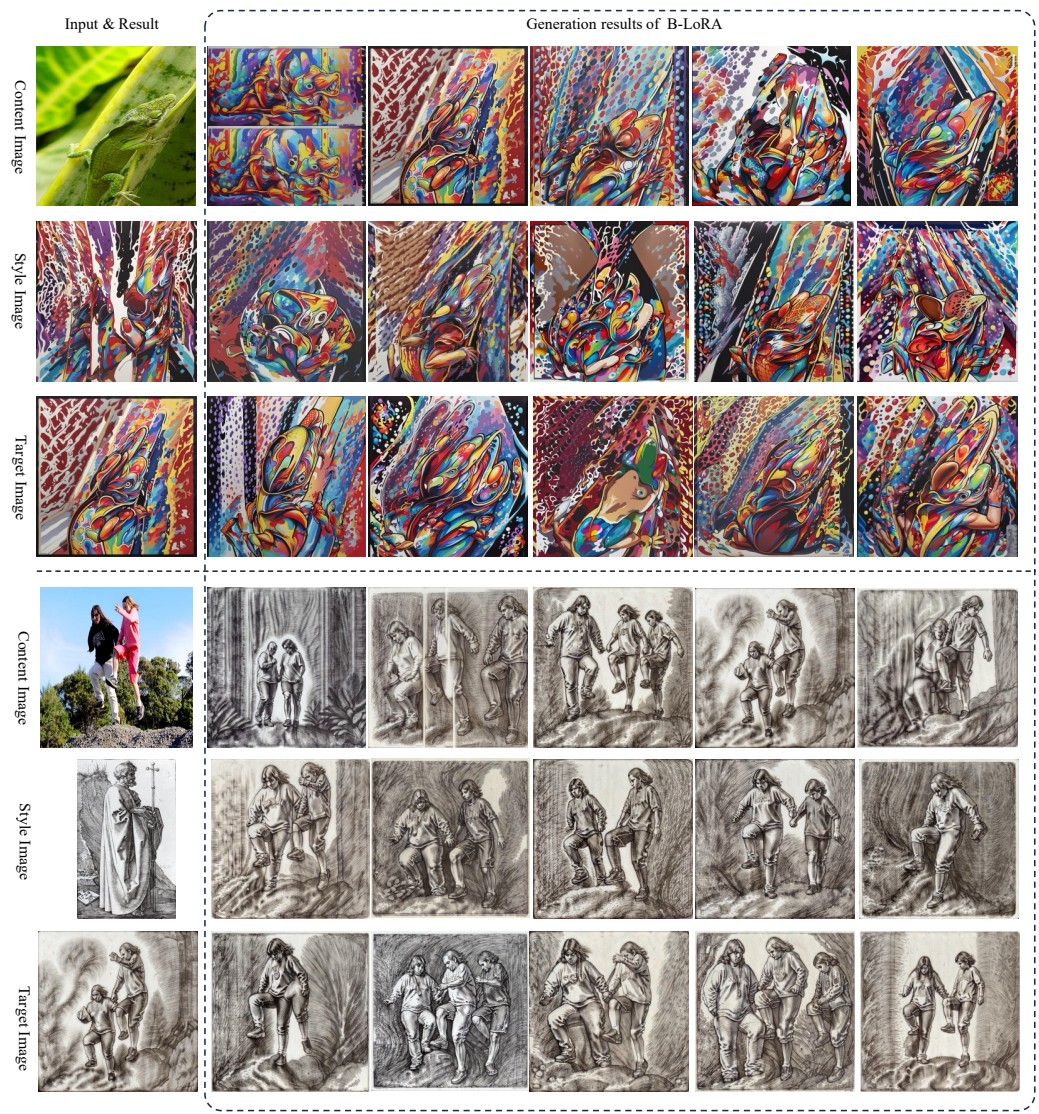

Figure 20: Example of data cleansing using CAS.

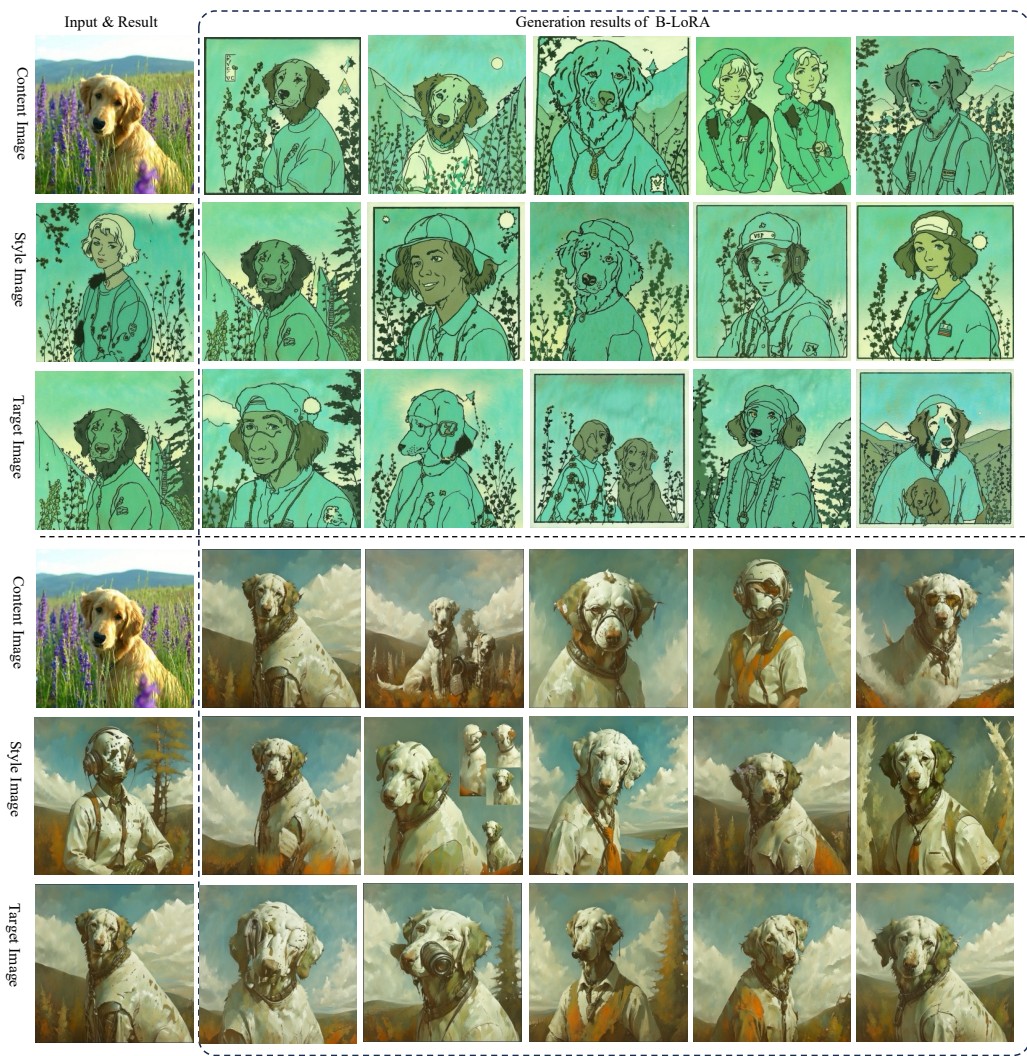

Figure 21: Example of data cleansing using CAS.

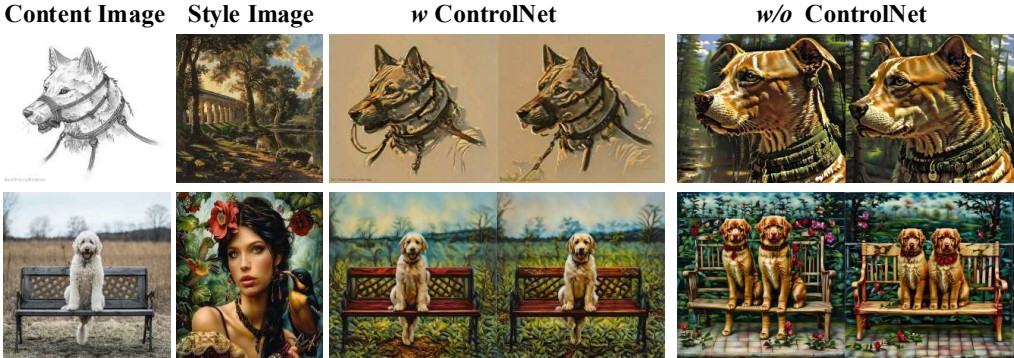

Figure 22: Ablation studies of ControlNet.

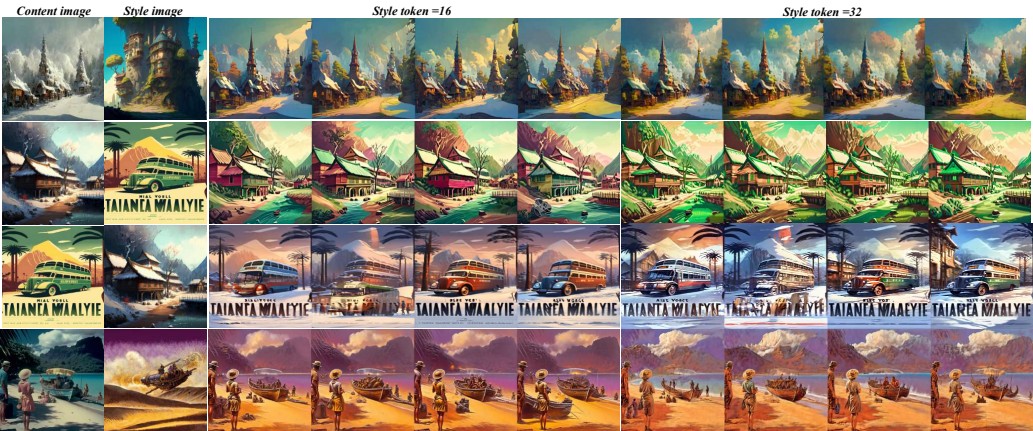

Figure 23: Ablation studies on style token.

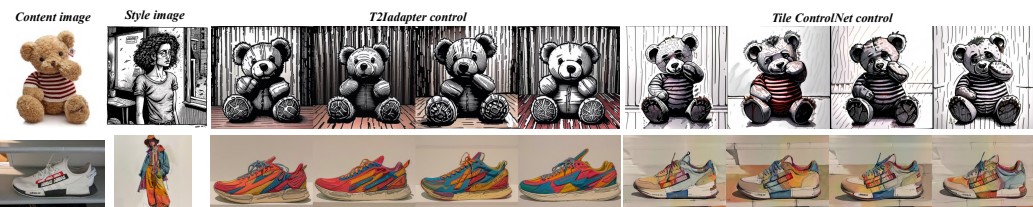

Figure 24: Results comparison between using line control content and original image control content.

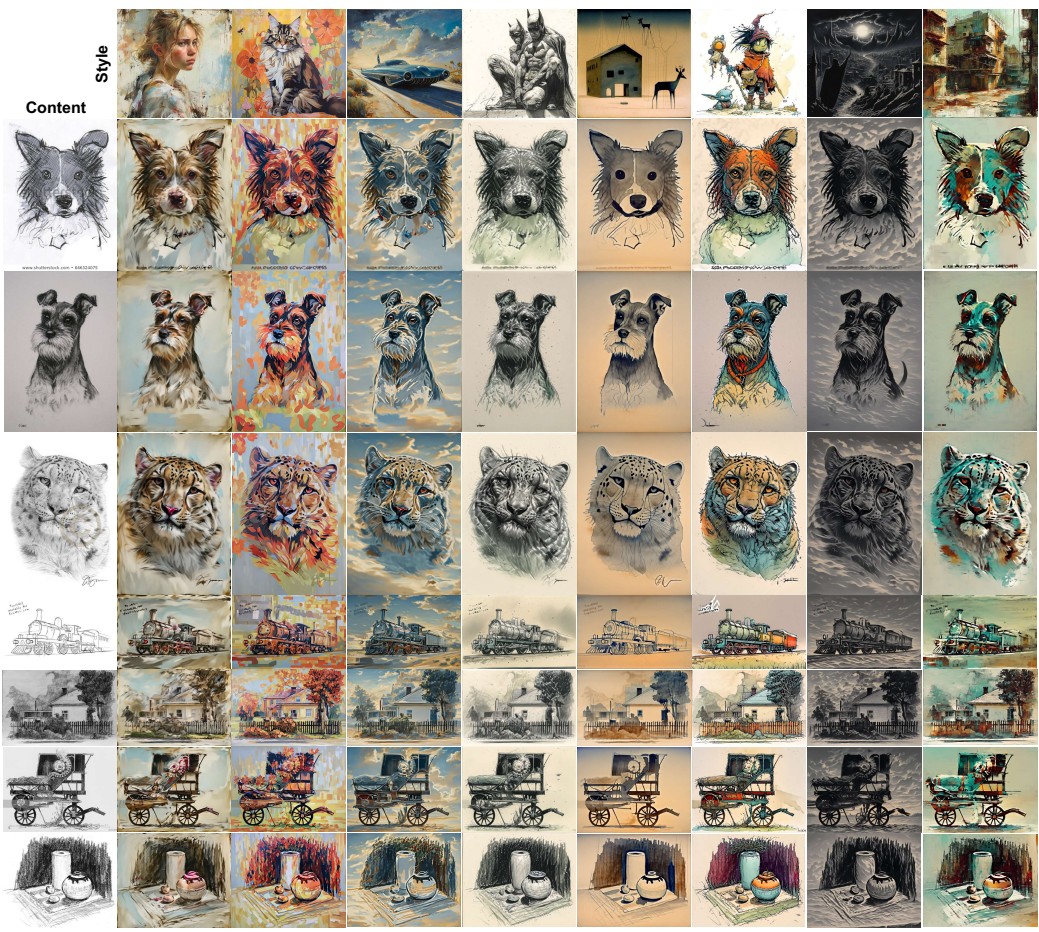

Figure 25: Sketch-driven style transfer results.

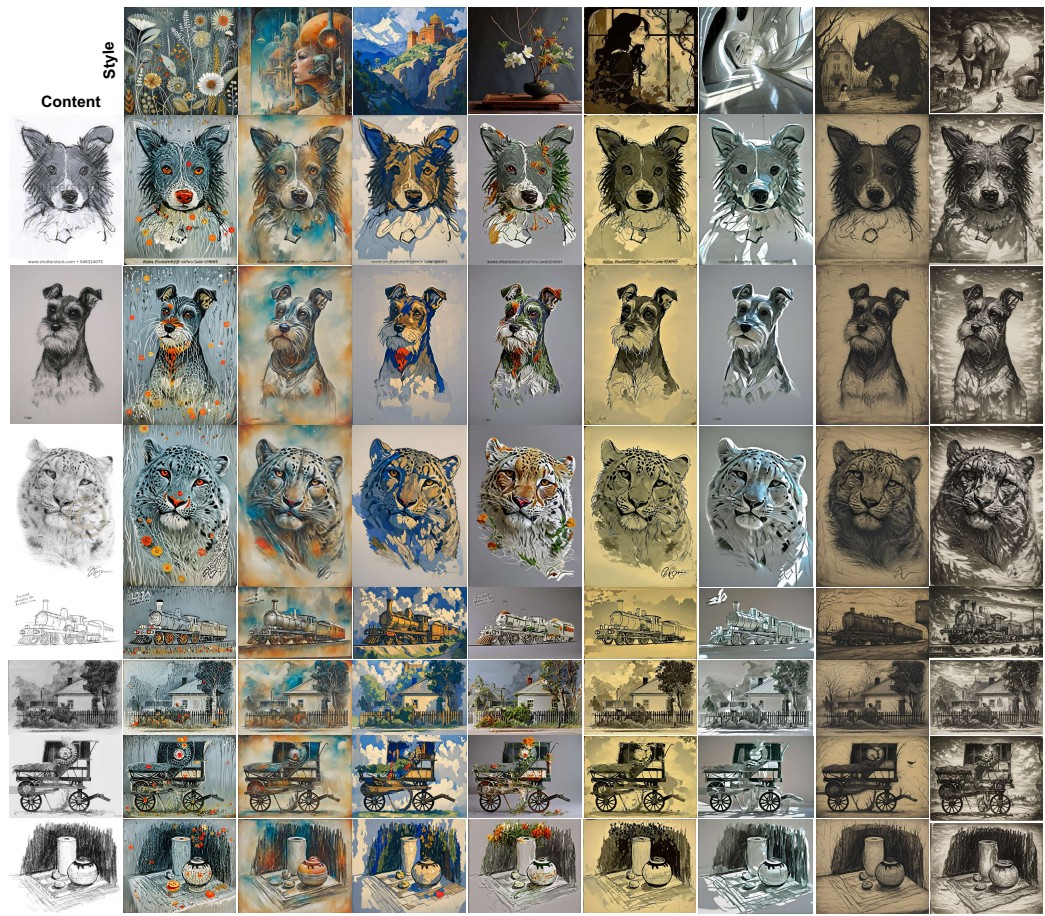

Figure 26: Sketch-driven style transfer results.

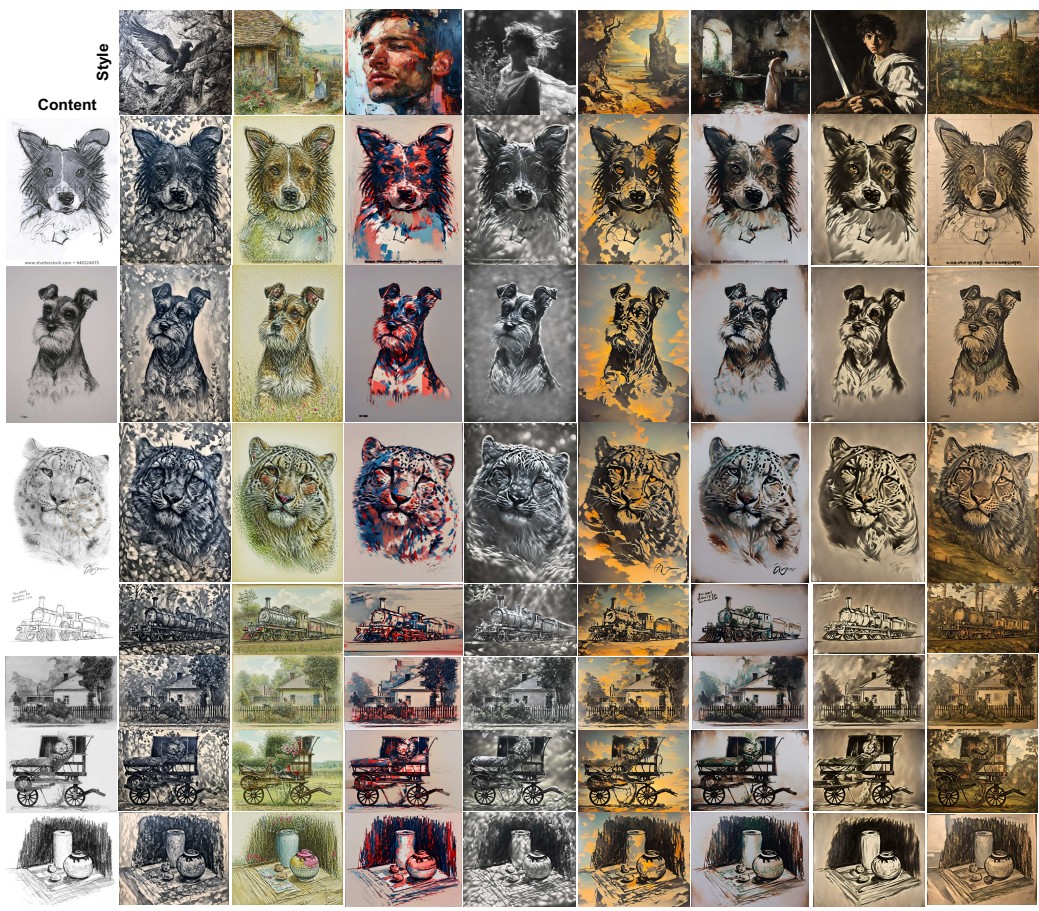

Figure 27: Sketch-driven style transfer results.

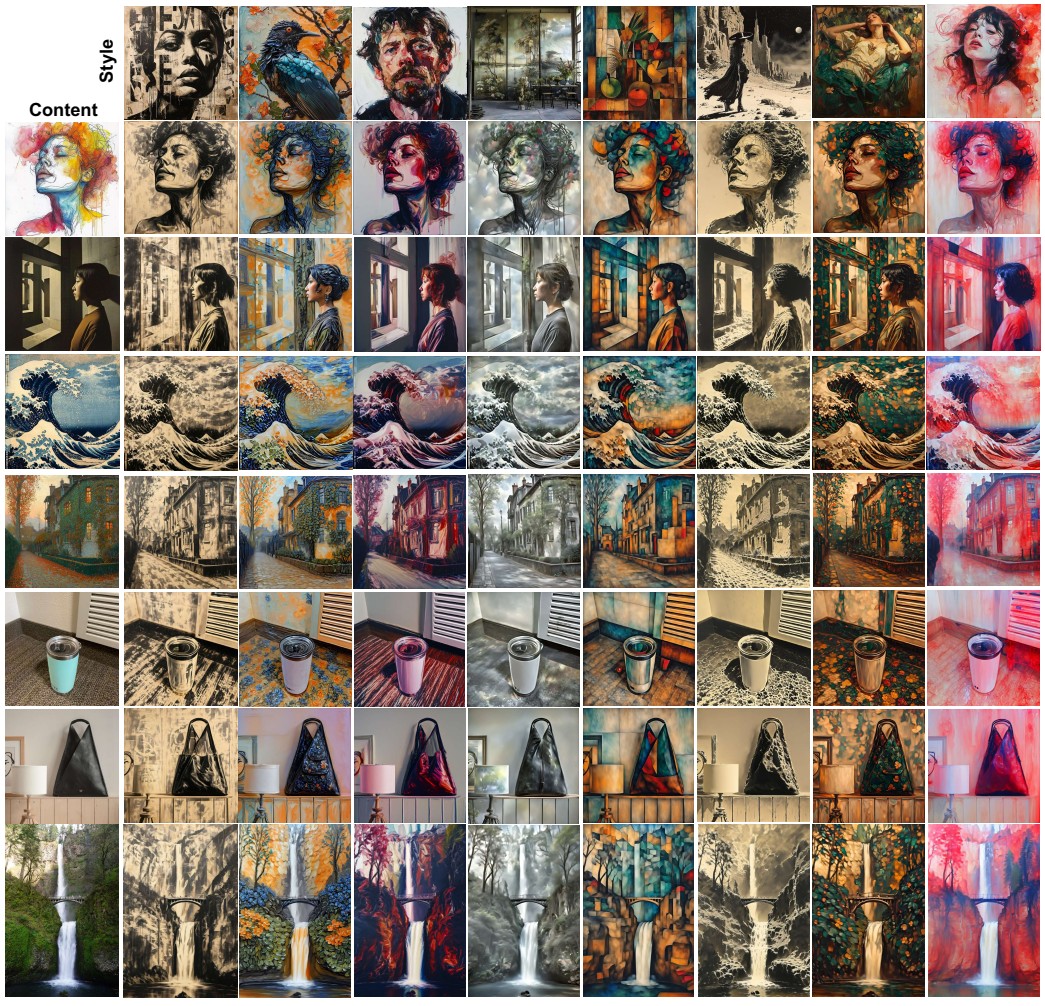

Figure 28: Image-driven style transfer results.

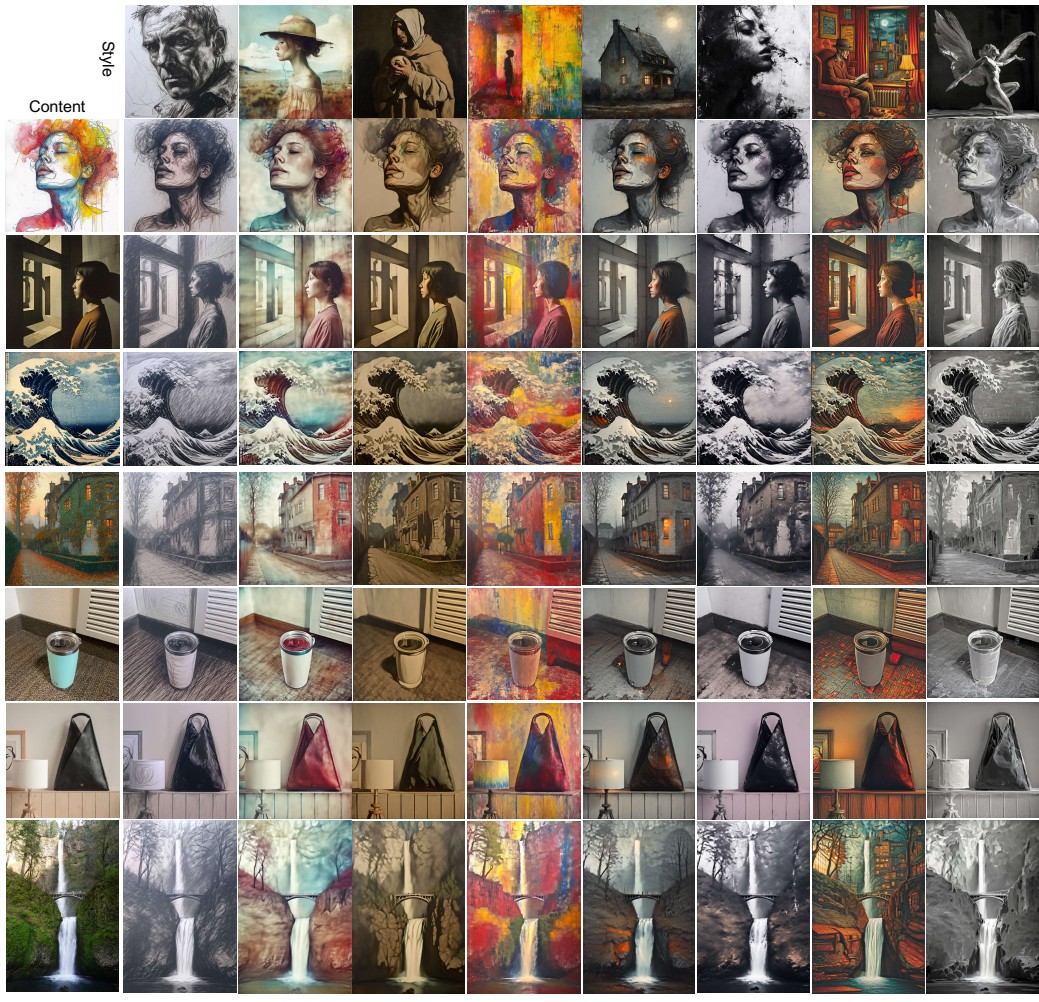

Figure 29: Image-driven style transfer results.

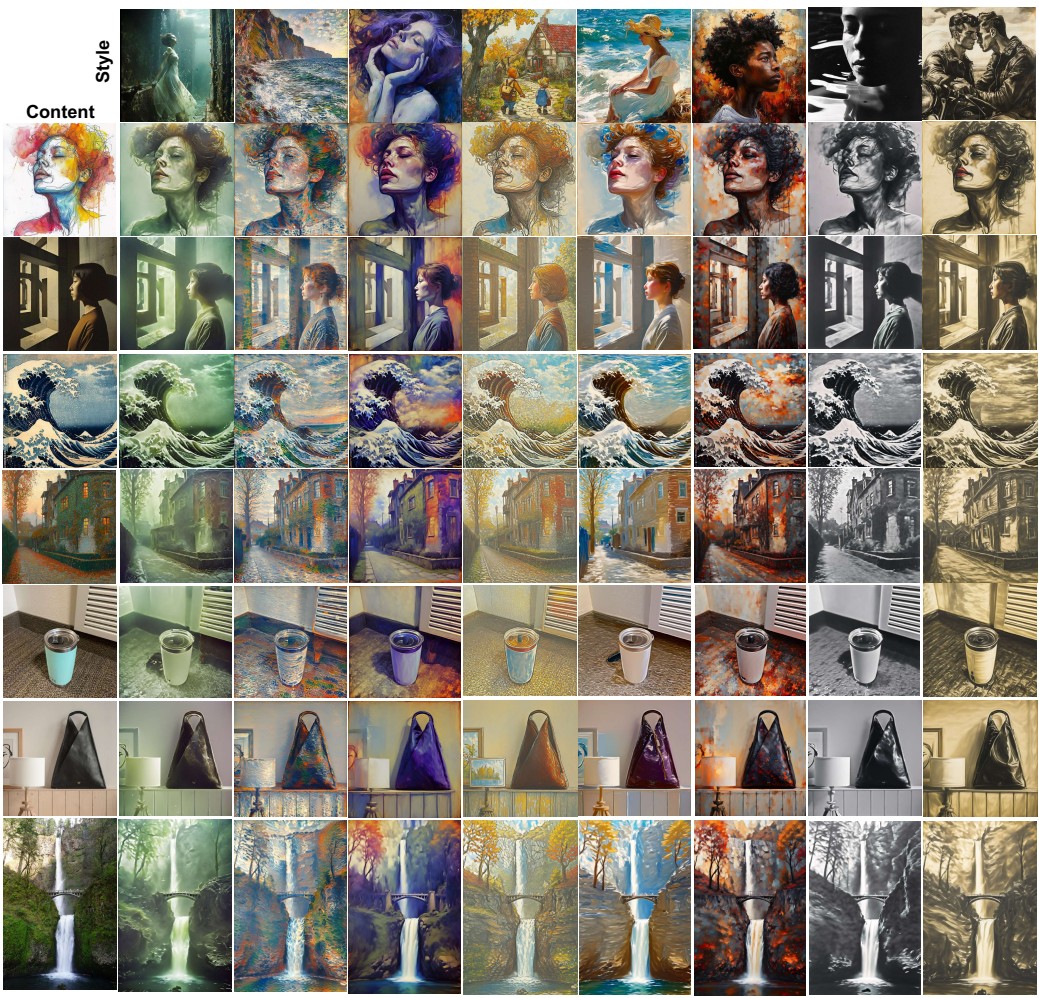

Figure 30: Image-driven style transfer results.

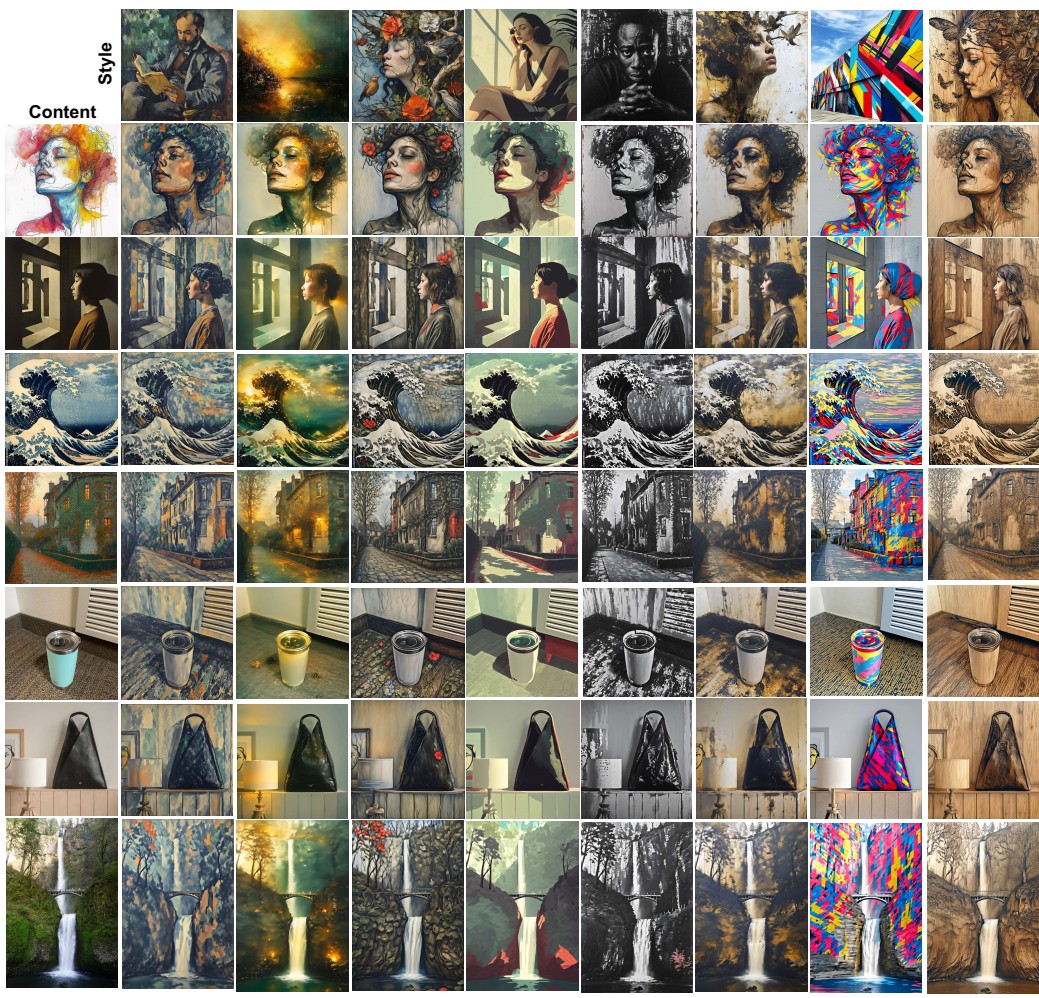

Figure 31: Image-driven style transfer results.

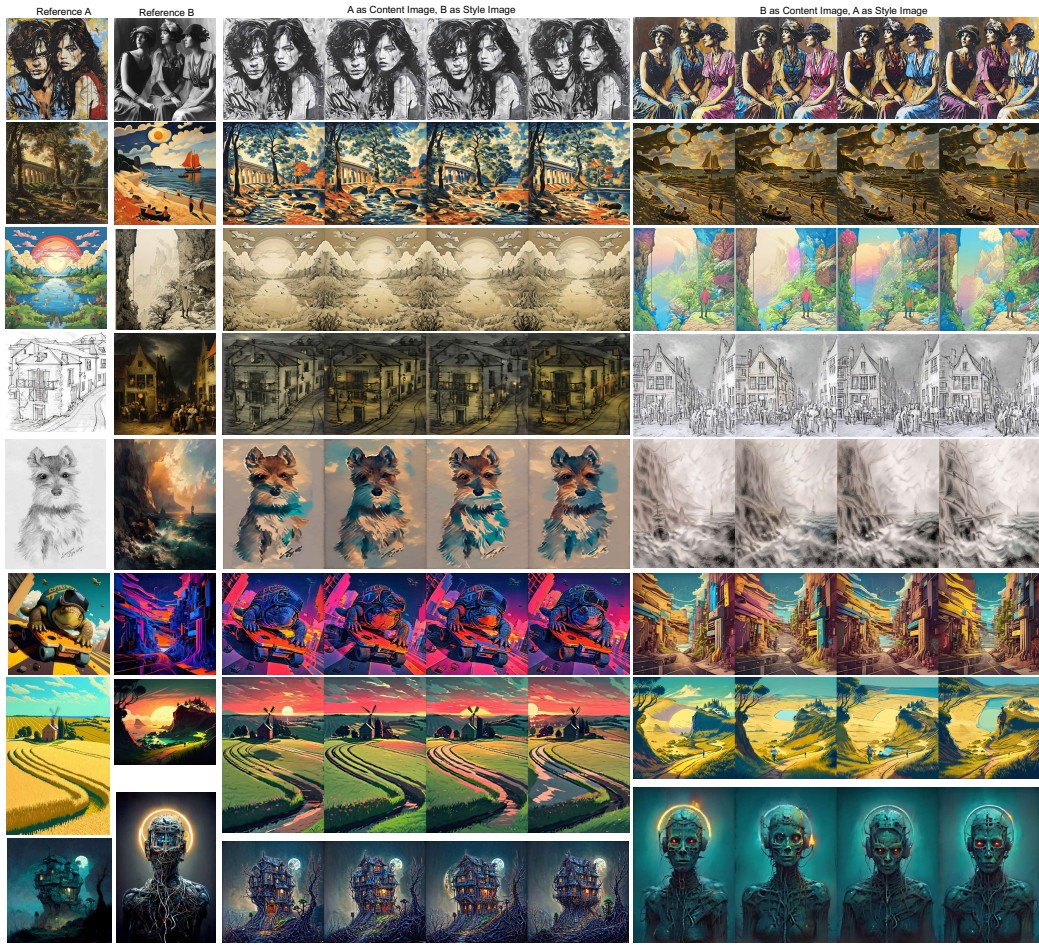

Figure 32: Content-style cycle transfer results.

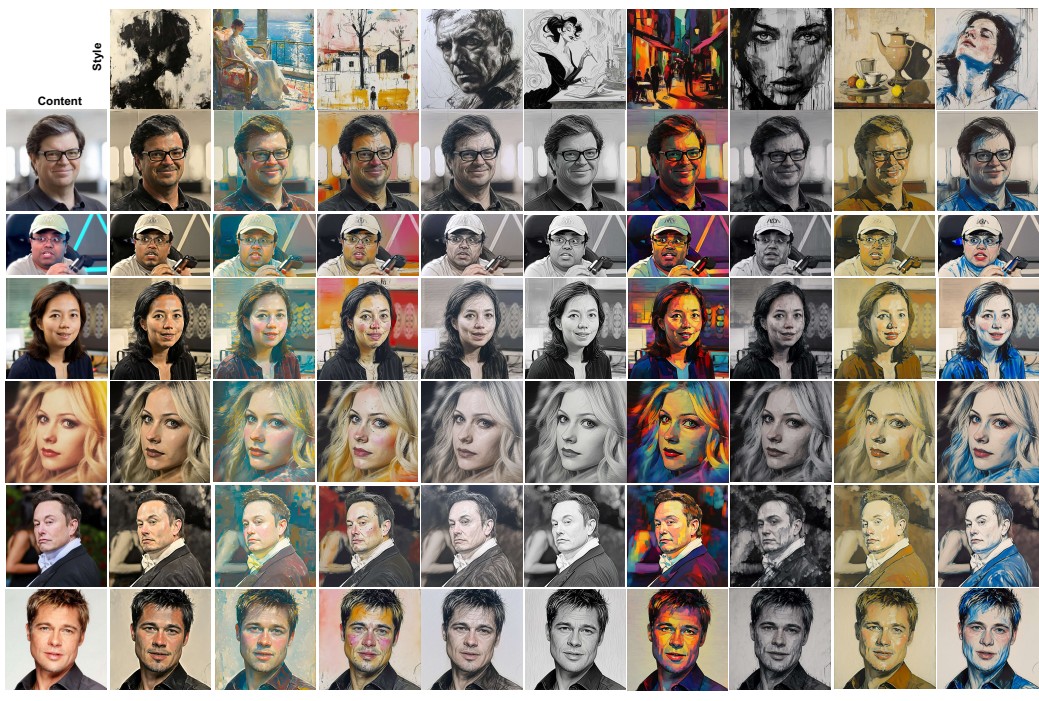

Figure 33: Face image style transfer results.

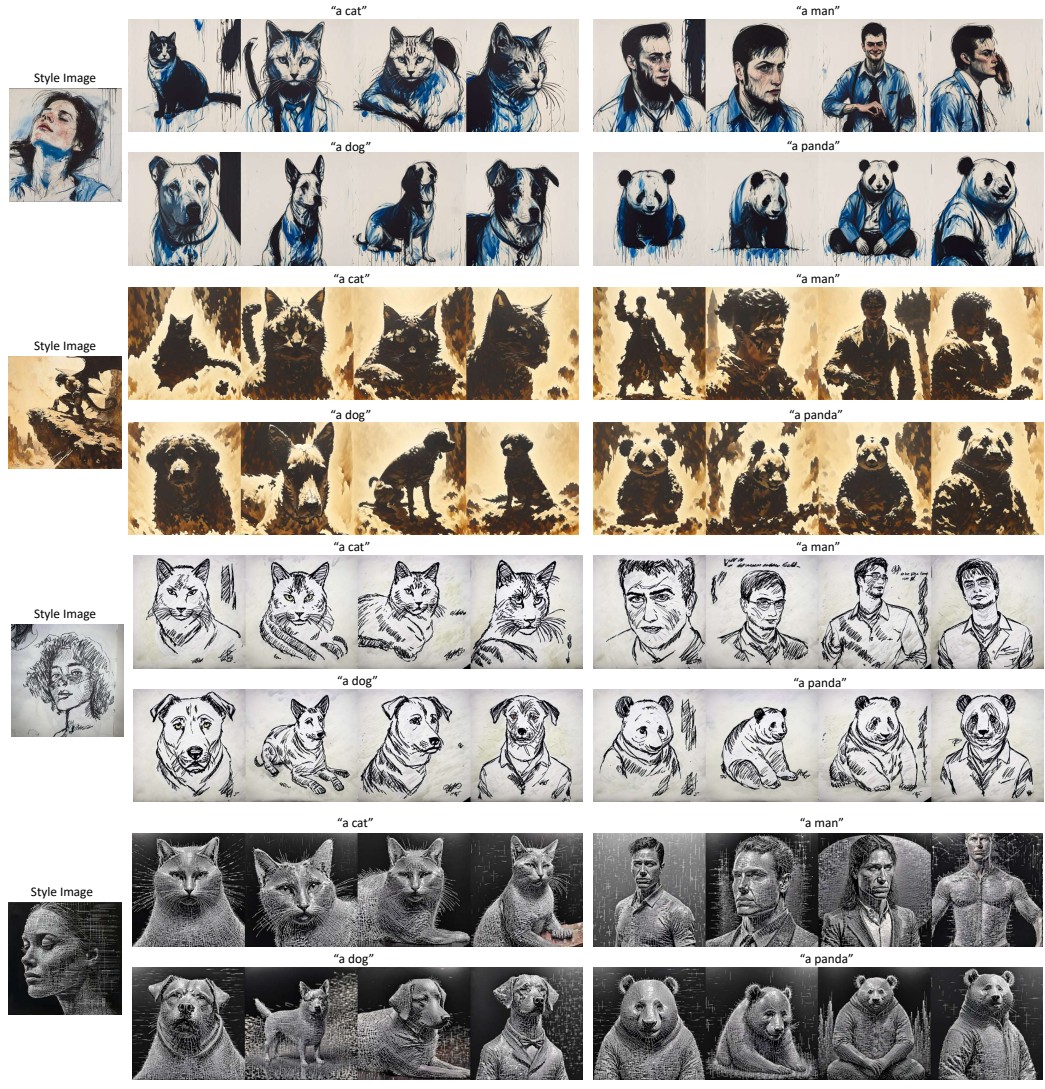

Figure 34: Text-driven stylized synthesis results.

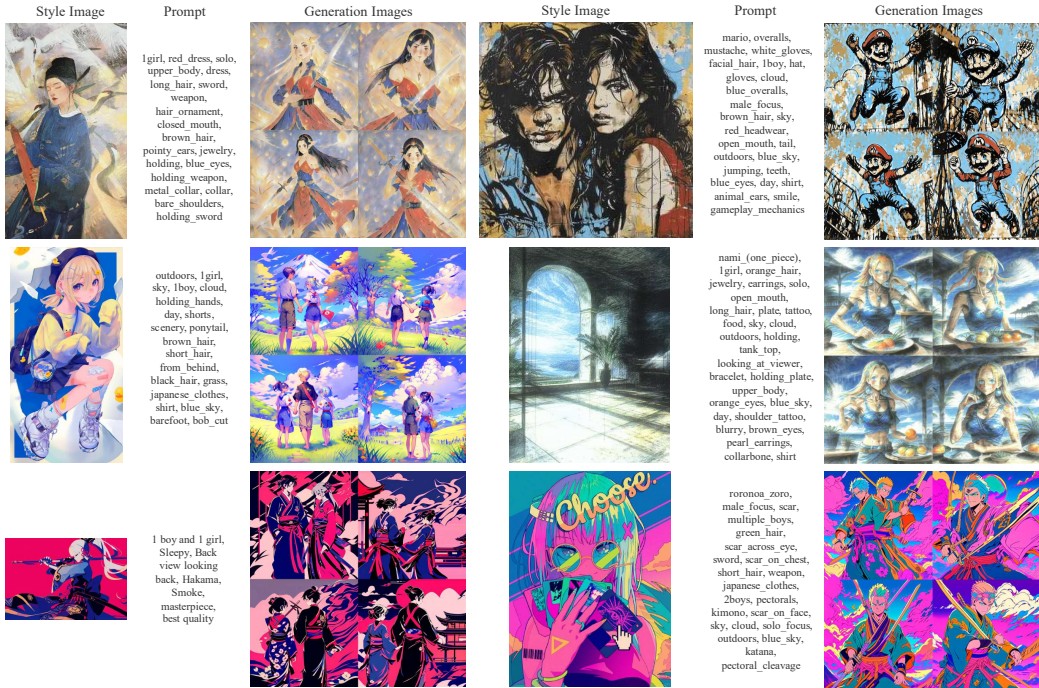

Figure 35: Text-driven stylized synthesis results.

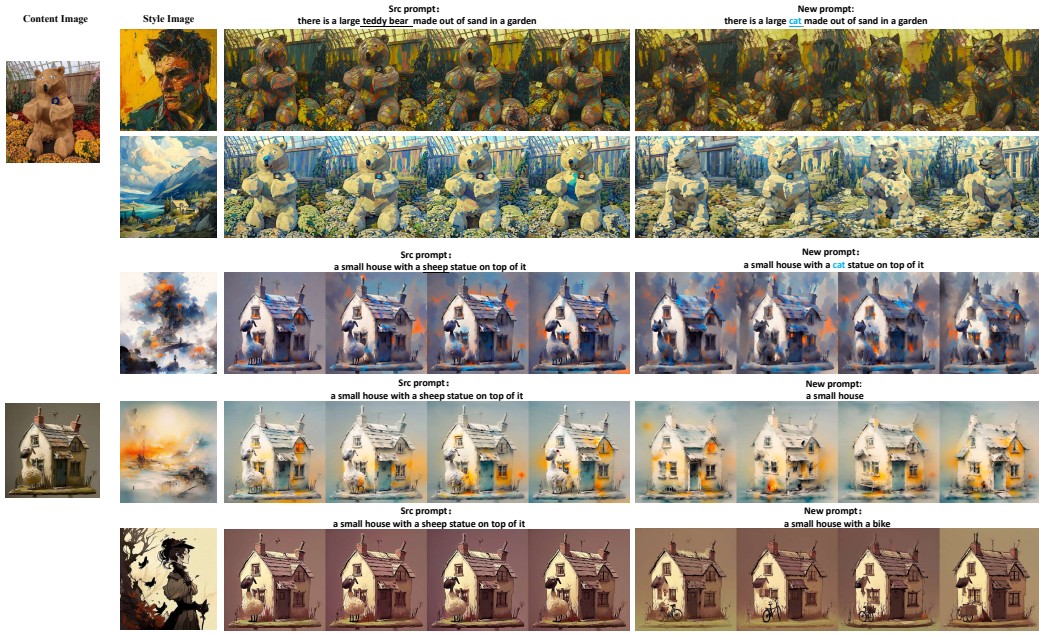

Figure 36: Text editing-driven stylized synthesis results.

