# OpenReview forum: "CSGO: Content-Style Composition in Text-to-Image Generation"
_NeurIPS.cc/2025/Conference — NeurIPS 2025 poster_

### Official Review · Reviewer_a2Ei · 2025-06-26

**Clarity:** 2
**Significance:** 3
**Originality:** 2
**Rating:** 4
**Confidence:** 3

**Summary:**

This paper establishes a new dataset, IMAGStyle, for style transfer with a <content, style, target> triplet structure. To obtain large-scale content-style-stylized image triplets, this paper introduces an automatic pipeline with a generation stage and a cleaning stage. This paper also proposes a style transfer approach that uses the IMAGStyle dataset for retrieval training. Experiments are widely conducted and show effectiveness in various style transfer scenarios.

**Questions:**

1. Although I appreciate the authors' efforts to construct such a large-scale dataset for style transfer, I doubt the necessity and quality of ground-truth stylized images. The task of style transfer is inherently subjective. It is generative and artistic, not reconstructive. On the other hand, the filtering algorithm (i.e., CAS) only relies on the content preservation metric, while the style information is not taken into consideration. This design might cause some content invasion from the style image. As shown in the examples of the first row in Fig. 2, two stylized dog images show the content of style images, i.e., wearing a shirt. This would hinder the disentanglement of content/style during training. This can also be observed in Fig. 4, a dandelion appears in the background of CSGO's result, which is the content of the style image. I guess this is also one reason for the failures of CSGO on portrait stylization (Appendix 2.1). Could the authors explain the reason for not introducing a style metric during data collection?

2. CSGO introduces several hyperparameters to control the scale of injected content or style. I found the results in Fig. 11 lack interpretability, e.g., determining how much to preserve structure or how much to reflect style. For example, the results of $\lambda_s=0.5,1,1.5$ all bias to the content image (might be because of a large $\delta_c=0.5$). Also, too much weights complicate this control process.

3. The architecture of CSGO is dedicated, and the ablation study design is not good. Since CSGO injects content features via (1) ControlNet, and (2) learnable cross-attention layer, what does "W/O Content Control" refer to? According to Fig. 3 (both in the box of the content control), it seems that two modules are both removed; in this case, the effect of each module is not fully studied. One suggestion to conceptualize these modules: (1) Content control, i.e., cross-attention in the down-sample blocks; (2) Style control,  i.e., cross-attention in the up-sample blocks; (3) Dual/Mixed control, i.e., ControlNet with information from both pixel-level content image and semantic-level style image. This might be clearer and easier for readers to follow, and help to design ablation experiments.

4. Regarding the style stream, I suggest the authors provide more details about the style control, e.g., style token (Perceiver Resampler), and the formulation of $\lambda_s$ (line 177). Also, it is not clear whether the input of the two cross-attention in style control is the same, i.e., both use F(S)' or two independent projections.

5. Most ablation experiments are visually studied, but lack quantitative results (either learning-based or human-based). Especially these scale parameters, a quantitative comparison may help to demonstrate the style-content trade-off.

**Ethical Concerns:**

["NO or VERY MINOR ethics concerns only"]

**Final Justification:**

The major contribution of this paper is the IMAGStyle dataset, which is constructed based on B-LoRA with CAS score filtering to ensure both style and content quality in the stylized images. This design is supported by the hypothesis and empirical findings that B-LoRA achieves style alignment but lacks content fidelity. The proposed CSGO method leverages this dataset and shows improvements over state-of-the-art methods towards controllable style transfer. While some concerns remain regarding the dataset quality and architectural design, I view the dataset as a positive contribution to the style transfer community and recommend a score of borderline accept.

**Limitations:**

The authors have discussed the limitations of CSGO in facial stylization scenarios, but the quality of this paper could be further improved with discussions on other failure cases (e.g., content leakage or content overfitting) and social impact.

**Quality:**

3

**Strengths And Weaknesses:**

**Strengths**:
1. This paper introduces a large-scale style transfer dataset, IMAGStyle, for stylized image retrieval.
2. A learning-based style transfer approach is proposed by leveraging the IMAGStyle dataset.
3. Experiments and ablation studies are extensively conducted to evaluate the proposed CSGO method.

**Weaknesses**:

1. The dataset largely relies on a well-established style transfer method B-LoRA and may involve several issues, such as a lack of generalization and adaptability, or content overfitting.
2. The model is architecturally complex, incorporating multiple specialized modules. It also introduces many hyperparameters that may complicate the generative controlling process. The efficiency of training and inference is not reported.
3. The experiments mostly focus on visual examples, but having more quantitative and statistically significant experiments could strengthen the paper's contributions further.

---

> ### Author Rebuttal · Authors · 2025-07-31
>
> Response to Reviewer a2Ei
>
> Thank you for the reviewers' recognition.
>  ***
> **Q1-1.** Why not introduce a style metric during data collection?
>
> **A1-1:** To address these concerns, we provide the following clarifications:
> (1) B-LoRA enables efficient and stable style transfer, consistently generating images that closely align with the style characteristics of the original references. This claim is substantiated by the following experimental validation:
>  We constructed 300 unique content-style LoRA combinations by pairing 300 content LoRAs with 300 style LoRAs. For each combination, we generated 15 stylized images. Style fidelity was rigorously quantified using three key metrics:
> (a) The mean style similarity between generated images and original style references,
> (b) The average maximum style similarity score per content-style group, and
> (c) The average minimum style similarity score per content-style group.
> To ensure robustness in style assessment, we employed two distinct style evaluation models (CSD and OneIG-StyleEncoder). Both models output similarity scores normalized to the range [-1, 1], where higher values indicate greater style congruence. As shown in the table below, the results confirm that B-LoRA effectively captures stylistic representations and achieves stable style transfer.
>
> |Metric|mean|max|min|
> |-|-|-|-|
> |**CSD ↑**|0.618|0.738|0.547|
> |**OneIG-styleEncoder ↑**|0.679|0.815|0.611|
>
> (2) The observed instability in B-LoRA stems primarily from content preservation inconsistencies rather than style transfer limitations. This distinction was verified through Content Alignment Score (CAS) analysis between generated images and original content references. We report three key metrics:
> (a) The mean CAS across all generated images,
> (b) The average maximum CAS per content-style group, and
> (c) The average minimum CAS per content-style group.
> The CAS metric (range: [0, ∞)) employs an inverse scale where lower values indicate superior content preservation. As shown in the table below, CAS values exhibit significant variability, confirming B-LoRA's susceptibility to content degradation.
>
> |Metric|mean|max-mean|min-mean|
> |-|-|-|-|
> |**CAS ↓**|1.67|2.43|1.13|
>
> (3) To further validate the robustness of our dataset, we randomly sampled 1000 triplets and computed the average CAS, CSD, and OneIG-StyleEncoder scores of the selected images. The experimental results confirm that the samples obtained through our selection strategy are both reliable and stable.
>
> |Metric|mean|
> |-|-|
> |**CSD ↑**|0.646|
> |**OneIG-styleEncoder ↑**|0.661|
> |**CAS ↓**|1.11|
>
> （4） The screening results using only CAS are more aligned with the manual selection results. To evaluate the alignment rare (AR) between different screening strategies and human selections, we conducted some verification experiments. We introduced a\*CAS + b*(1-CSD) to represent the final score used for screening. We filtered the results using four strategies respectively and calculated the alignment with the manually screened results (top 1). It can be observed that due to the instability in the content LoRA of B-LoRA, the results screened using only CAS are actually the closest to the manual screening results. Therefore, we choose to screen the dataset solely based on the CAS score.
>
> |User|a|b|AR|User|a|b|AR|User|a|b|AR|User|a|b|AR|User|a|b|AR|
> |-|-|-|-|-|-|-|-|-|-|-|-|-|-|-|-|-|-|-|-
> |A|1.0|0|85.3%|A|0.75|0.25|65.3%|A|0.5|0.5|53.3%|A|0.25|0.75|16.7%|A|0|1.0|9.7%|
> |B|1.0|0|84.0%|B|0.75|0.25|59.3%|B|0.5|0.5|50.3%|B|0.25|0.75|18.3%|B|0|1.0|10.7%|
> |C|1.0|0|82.7%|C|0.75|0.25|58.3%|C|0.5|0.5|47.7%|C|0.25|0.75|15.3%|C|0|1.0|13.0%|
> |D|1.0|0|86.7%|D|0.75|0.25|55.3%|D|0.5|0.5|40.0%|D|0.25|0.75|18.3%|D|0|1.0|13.3%|
>
>  ***
> **Q1-2.** Explanation of the "Dandelion" in Figure 4.
>
>  **A1-2:** Comparative analysis in Figure 4 demonstrates that our method achieves superior stylistic alignment while maintaining optimal content preservation. We posit that stylistic representation inherently encompasses perceptual subjectivity, where brushstroke techniques, color palettes, and compositional arrangements constitute fundamental stylistic components. Illustratively, in a feline-form painting composed of "grass" elements, these botanical features function dually as content carriers and stylization primitives. This ontological duality suggests that strategically incorporating content-distinct elements can enhance style expression in transfer tasks.
>
>
> ***
> **Q2.** Concerns about the hyperparameters.
>
> **A2:** Beyond the established Classifier-Free Guidance (CFG) technique commonly employed in diffusion models, the proposed CSGO method introduces two critical hyperparameters: the content scale δ_c and the style scale λs. Analogous to conditioning mechanisms in models such as Adapter or ControlNet, these hyperparameters are designed to enable finer-grained control over generative outcomes during inference.
> As illustrated in Figure 11 and Figure 12, we demonstrate the impact of all hyperparameters on style transfer results. The presented examples show that these hyperparameters provide stable and controllable influence: for instance, a CFG value of 10 yields optimal performance in both example sets, while a content scale δc of 0.5 also achieves the best results in both cases.
> Given the inherently subjective nature of style perception, these hyperparameters facilitate adjustable generation of style transfer outputs tailored to specific requirements, thereby accommodating the need for diverse stylistic interpretations. Crucially, these hyperparameters are not unduly complex or difficult to tune; their design aligns with an intuitive understanding of the style transfer task, ensuring both accessibility and robust control. The ease of use has been systematically validated in our experiments.
>
>
> **Table 1. Ablation studies of \δ_c.**
>
> |Metric|0.3|0.5|0.7|0.9|
> |-|-|-|-|-|
> |**CSD ↑**|0.522|0.515|0.499|0.402|
> |**CAS ↓**|0.964|0.839|0.820|0.797|
>
> **Table 2. Ablation studies of λs.**
> |Metric|0.5|1.0|1.5|
> |-|-|-|-|
> |**CSD ↑**|0.406|0.515|0.554|
> |**CAS ↓**|0.784|0.839|0.883|
>
> ***
>
> **Q3.** Results of the new ablation experiments.
>
> **A3:** We thank the reviewers for their insightful suggestions. In response, we refined our ablation study to systematically evaluate three critical components: (1) the content control module (cross-attention layers in downsampling blocks), (2) the style control module (cross-attention layers in upsampling blocks), and (3) the dual-control fusion strategy (the style features are inject to Controlnet ).(4) the dual-control fusion strategy without style feature. Quantitative results validating each module's contribution are presented as follow. Obviously, the hybrid content control module is missing, resulting in weak content feature retention capability. The style features are mainly related to the upsampling style control module and the style injection of the Controlnet branch. In order to balance the migration capabilities of style and content, we finally integrated the above-mentioned modules to achieve the optimal results.
>
>
> |Metric|w/o (1)|w/o (2)|w/o (3)|w/o (4)|CSGO|
> |-|-|-|-|-|-|
> |**CSD ↑**|0.512|0.431|0.546|0.509|0.515|
> |**CAS ↓**|0.852|0.847|1.578|0.846|0.839|
>
> ***
> **Q4.** Please provide more details about the style token (Perceiver Resampler), as well as the formulation of (line 177) and F(S)' in style control.
>
> **A4:** (1) The style token refers to the number of tokens obtained after mapping the original style features using a resampler structure. A larger number of tokens means that more style features are retained.
> （2）Line 177 mainly describes that we not only inject style features into the upsampling layer, but also inject them into the Controlnet control module through an independent cross-attention module.
> （3） Yes, we use the same F(S)', but with two independent A.
>
> ***
> **Q5.** A quantitative comparison may help to demonstrate the style-content trade-off.
>
> **A5:** We thank the reviewers for their constructive feedback. In addition to the previously reported quantitative ablation studies on content- and style-related hyperparameters , we have supplemented the following:
> (1) Quantitative results analyzing the impact of style token t,
>
> |t|4|12|16|32|
> |-|-|-|-|-|
> |**CSD ↑**|0.4828|0.5046|0.5017|0.5146|
> |**CAS ↓**|0.8275|0.8312|0.8428|0.8386|
>
> (2) Performance metrics for text-driven stylized synthesis tasks
>
> |method|InstantStyle|StyleShot|StyleAligned|DEADiff|CSGO(ours)|
> |-|-|-|-|-|-|
> |**CSD ↑**|0.613|0.632|0.497|0.485|**0.648**|
> |**CLIP-Score ↑**|0.295|0.292|0.289|0.287|**0.309**|
>
>
> Collectively, these comprehensive quantitative analyses—supported by our visualization results—demonstrate CSGO's robustness in style transfer applications. We believe this expanded evidence addresses the reviewers' methodological concerns regarding the approach's validity.
>
>
>
>
> ***
> **Q6.** inference time and training efficiency.
>
> **A6:** Regarding the training efficiency of B-LoRA: On a single H800, it only takes 6 minutes to train a B-LoRA. Regarding the inference efficiency of CSGO: On a single H800, without any optimization measures, it takes 4.5 seconds at 512 resolution and 12 seconds at 1024 resolution to generate results. In fact, the inference time of CSGO is significantly lower than that of other style transfer models. Most other stylization methods rely on DDIM inversion, which almost requires doubling the time for DDIM inversion.
>
>  ***
>  **Q7.** The authors have discussed the limitations of other failure cases and social impact.
>
> **A7:** Thank you for the reviewers' suggestions. In CSGO, due to the introduction of Controlnet, the content signal is very strong, and unreasonable inference hyperparameters may lead to excessive content, that is, style loss; setting the content control hyperparameter to 0.5 will help. Our research is the first to verify that large-scale synthetic data can improve style transfer, providing a new baseline, which will be helpful to the community.

---

> > ### Comment · Reviewer_a2Ei · 2025-08-03
> > **Response to rebuttal**
> >
> > Thanks for the rebuttal. I appreciate the authors' engagement in addressing all reviewers' concerns. My questions about hyperparameters (Q2) and architecture designs (Q3) have been well addressed. These results validate the controllability of CSGO in the content-style trade-off for practical use.
> >
> > The shared concern between the reviewers is the **IMAGStyle** dataset. I appreciate the step-by-step clarification in the rebuttal. The results indeed support the authors' hypothesis that B-LoRA achieves style fidelity but lacks content preservation, thereby validating the choice of solely using CAS for dataset construction. However, from A1-1 (3), the authors state "sampled 1000 triplets and computed the average CAS, CSD, and OneIG-StyleEncoder scores of the selected images", I would like to know more details about this. For example, given a triplet $(I_{content}, I_{style}, I_{stylized})$, is CAS (which assesses content alignment) computed between $(I_{content}, I_{stylized})$, and CSD/OneIG-StyleEncoder (which assesses style similarity) computed between $(I_{style}, I_{stylized})$?
> >
> > If these triplets are considered the **ground truth** of this dataset, does this result represent an **upper bound** for performance? Since a CSD score of 0.5 can indicate the absence of the style [1], the GT's score of 0.646 suggests less-than-ideal stylization. Another observation: compared to Table 1, where CSGO achieves (CSD: 0.5146, CAS:0.8386), I guess CSGO with ControlNet (the authors claim the content signal in ControlNet is very strong) enhances the content but compromises the style. Also, it is not bad to acknowledge a limitation in CSGO regarding the redundant content from the style image, as the quantitative results are convincing and support its superiority in style alignment (Table 1, best CSD of CSGO).
> >
> > I remain open to further discussion with the authors and other reviewers.
> >
> > [1] Measuring Style Similarity in Diffusion Models

---

> > > ### Author Response · Authors · 2025-08-04
> > > **Response to Reviewer a2Ei**
> > >
> > > We thank the reviewers for their careful assessment. We welcome the opportunity to discuss further details regarding CSGO. Below, we address the reviewers' concerns point by point.
> > >
> > > ***
> > >
> > > **Q1.** Details on the calculation of CAS and CSD for randomly sampled triplets.
> > >
> > > **A1:** As correctly noted by the reviewer, for a given triplet (I_content, I_style, I_stylized), the Content Alignment Score (CAS) is computed between images I_content and I_stylized, while the Style Similarity Score (CSD) is computed between images I_style and I_stylized.
> > >
> > > ***
> > > **Q2.** Further explanation regarding the value range of CSD.
> > >
> > > **A2:** In fact, both CSD and OneIGStyleEncoder calculate the cosine similarity between vectors, and their actual value range is [-1, 1]. Therefore, a style similarity of around 0.646 already indicates a perceptually high level of style similarity.
> > >
> > > ***
> > > **Q3.** Do these results represent the upper bound of performance?
> > >
> > > **A3:** We believe that this cannot temporarily represent the upper bound of style transfer performance. In fact, for the initial large-scale training, we retained as many triplets with low CAS results as possible. There are some cases of poor quality within these, and the data quality can be further improved by cleaning the data and retaining triplets with low CAS and high CSD. In addition, as discussed in the fifth point with **Reviewer TeEm**, style transfer datasets have the conditions for rapid expansion, which can be achieved using B-LoRA and CSGO themselves. Our main contribution is to verify that large-scale synthetic style transfer datasets can enhance the effectiveness of style transfer tasks that are difficult to collect in reality. Therefore, when the data quality is improved, the performance of style transfer can theoretically be further enhanced.
> > >
> > > **Q4.** Another observation: compared to Table 1, where CSGO achieves (CSD: 0.5146, CAS:0.8386), I guess CSGO with ControlNet (the authors claim the content signal in ControlNet is very strong) enhances the content but compromises the style. Also, it is not bad to acknowledge a limitation in CSGO regarding the redundant content from the style image, as the quantitative results are convincing and support its superiority in style alignment (Table 1, best CSD of CSGO).
> > >
> > > **A4:** In CSGO, we inject styles into the upsampling module and ControlNet. This is based on the observation that "the content signal in ControlNet is very strong) enhances the content but compromises the style.". Therefore, when style features are injected into both ControlNet and the upsampling module, the style transfer capability of CSGO is further improved. Finally, we appreciate the reviewer's recognition of CSGO's performance. We will further explore more advanced solutions for style transfer.
> > > ***
> > > If the reviewer has any questions, please feel free to add comments.

---

> ### Author Response · Authors · 2025-08-05
> **Response**
>
> Dear Reviewer a2Ei,
>
> We sincerely thank you again for reviewing our response. We are keen to confirm whether the answers and experiments provided have sufficiently addressed the relevant questions and concerns. Please do not hesitate to raise any further inquiries you may have. Thank you sincerely for your dedicated efforts and ongoing support.
>
> With kind regards, The CSGO Author Team

---

> > ### Comment · Reviewer_a2Ei · 2025-08-05
> >
> > Thanks for the rebuttal. I agree that the data filtering can be used to improve the data quality. Most of my concerns regarding this dataset have been addressed. I would like to raise my score from "borderline reject" to "borderline accept".

---

> > > ### Author Response · Authors · 2025-08-05
> > > **Response to  Reviewer a2Ei**
> > >
> > > Dear Reviewer a2Ei,
> > >
> > > We sincerely appreciate your recognition, which is very encouraging.  Should you have any further questions, please do not hesitate to reach out. Once again, we sincerely express our gratitude for your dedicated efforts and continued support.
> > >
> > > With kind regards, The CSGO Author Team

---

### Official Review · Reviewer_TeEm · 2025-06-29

**Clarity:** 2
**Significance:** 3
**Originality:** 3
**Rating:** 5
**Confidence:** 2

**Summary:**

The authors propose IMAGStyle, a large-scale dataset of 210K content-style-stylized triplets, created using an automated pipeline to overcome the lack of structured data in style transfer. They further introduce CSGO, a unified framework that decouples content and style for versatile and high-quality style transfer. Experiments show state-of-the-art performance, emphasizing the value of structured data.

**Questions:**

Please refer to the weaknesses.

**Ethical Concerns:**

["NO or VERY MINOR ethics concerns only"]

**Final Justification:**

All concerns are addressed. Thanks!

**Quality:**

3

**Strengths And Weaknesses:**

[Strengths]
* The research topic regarding stylized image generation is important.
* Experiments validate the effectiveness of the proposed methods.

[Weaknesses]
* Some typos in the paper. For example, in line 97, loRA should be modified to LoRA. imges should be modifed to images.
* The current form of Figure 3 is a little bit messy. Maybe the authors should consider re-draw the pipleline to make it more readable.
* According to Figure 4, I can not find the strength of the proposed method. By contrast, the compared method is better regarding the generation quality, such as instantstyle.
* Authors mainly explore the data-driven way to achieve the stylized image generation. I am thinking about, if there is a large amount of data, the model architecture could be more simplified. For example, we can simply use the architecture of MLLM, such as Kosmos-G and Bagel to achieve this task.
* Could the authors explain more about the potential to expand the dataset, such as the possibility to scale the dataset to infinite amount of data. That could be more useful for the community.

---

> ### Author Rebuttal · Authors · 2025-07-31
>
> Response to Reviewer TeEm
>
>  We sincerely appreciate the reviewers' recognition of the strengths of our method. Next, we will respond to the reviewers' questions and concerns one by one.
> ***
> **Q1.** Some typos in the paper. For example, in line 97, loRA should be modified to LoRA. imges should be modifed to images.
>
> **A1:** Thank you for the careful review. We will correct any typos.
>
> ***
> **Q2.** The current form of Figure 3 is a little bit messy. Maybe the authors should consider re-draw the pipleline to make it more readable.
>
> **A2:** We thank reviewers for their valuable suggestion. We will implement the proposed architectural decomposition separating control mechanisms into: (1) content control in downsampling blocks, (2) style control in upsampling blocks, and (3) dual-control fusion. Due to current response constraints, this refinement will be implemented in the camera-ready manuscript.
>
>  ***
> **Q3.** According to Figure 4, I can not find the strength of the proposed method. By contrast, the compared method is better regarding the generation quality, such as instantstyle.
>
> **A3:** Thank you for your attention. In Figure 4, the advantages of our method over InstantStyle are as follows: In the first row, InstantStyle has the issue of "the trees next to the house being lost"; in the second row, it suffers from the loss of key facial attributes and insufficient style restoration; in the third row, InstantStyle loses background attributes; in the fourth row, its generated results lack the "stone texture" possessed by the style image.
> The reason for these differences is that InstantStyle relies on DDIM inversion based on content images to preserve content features, while DDIM inversion not only tends to lose content information of the original image but also takes almost twice as much time. Moreover, through large-scale training, the effectiveness of our proposed method in style transfer has been further improved.
>
>   ***
>
> **Q4.** Authors mainly explore the data-driven way to achieve the stylized image generation. I am thinking about, if there is a large amount of data, the model architecture could be more simplified. For example, we can simply use the architecture of MLLM, such as Kosmos-G and Bagel to achieve this task.
>
> **A4:** We thank the reviewers for their insightful observations. We concur that integrated generation-understanding architectures (e.g., Bagel, GPT-4o) represent an emerging paradigm where downstream task incorporation can streamline model design. While we acknowledge that sufficient data could theoretically enable Bagel to perform style transfer through full fine-tuning, such adaptation to multi-image inputs would necessitate substantial computational resources—a promising but resource-intensive research direction. Our plugin-based approach offers distinct advantages: it preserves base model capabilities while enabling parameter-efficient adaptation. We are actively exploring unified architectures for stylization and multimodal understanding-to-generation pipelines to bridge this methodological gap.
>
> In fact, our CSGO is still a basic and simple pipeline. We mainly verified that the automated large-scale construction of stylized datasets can significantly improve the effect of style transfer. We will actively explore a more elegant framework.
>
>   ***
>
> **Q5.** Could the authors explain more about the potential to expand the dataset, such as the possibility to scale the dataset to infinite amount of data. That could be more useful for the community.
>
> **A5:**
> The scalability of IMAGStyle is architecturally facilitated through an automated pipeline. By training m content-specific and n style-specific B-LoRAs, we theoretically obtain m × n triplets (content image, style image, stylized output). Each additional B-LoRA linearly extends the combinatorial space by m or n triplets respectively. Crucially, B-LoRA training exhibits high efficiency—requiring merely ≈6 minutes per module on NVIDIA H800 hardware. Thus, given sufficient content/style images and GPU resources, the system can rapidly scale to massive triplet volumes. Furthermore, CSGO serves as a complementary expansion mechanism: rather than training new B-LoRAs, it enables immediate generation of novel triplets through direct application, substantially accelerating dataset growth with minimal marginal cost.

---

> ### Author Response · Authors · 2025-08-05
> **Response**
>
> Dear Reviewer TeEm,
>
> We have proactively addressed the concerns raised by the reviewers and engaged in discussions regarding the dataset expansion and pipeline improvements of CSGO. We hope our responses will meet with your approval.
> We are keen to confirm whether the answers and experiments provided have sufficiently addressed the relevant questions and concerns. Please do not hesitate to raise any further inquiries you may have. Thank you sincerely for your dedicated efforts and ongoing support.
>
> With kind regards, The CSGO Author Team

---

### Official Review · Reviewer_nay9 · 2025-06-30

**Clarity:** 3
**Significance:** 3
**Originality:** 3
**Rating:** 5
**Confidence:** 4

**Summary:**

This paper was well written with clear structure and easy to understand. The paper firstly proposes a high quality and carefully cleaned dataset with 210k Content-Style-Stylized Image Triplets. Then, the paper proposes a new style transfer framework CSGO, which uses independent content and style feature injection modules to achieve high-quality image style transformations. Finally, a new score matrix named CAS was introduced to measure content loss after content-style transferred.

**Questions:**

A total of 11,000 content images and 10,000 style images would theoretically yield 110 million potential combinations, yet the experimental output amounts to only 210,000 images. What explains this notable discrepancy between the theoretical combinatorial possibility and the actual production yield?

Why injecting style features into ControlNet warrants a more detailed analysis.

Table 2 requires a more detailed explanation. Table 2 references appear to be missing.

**Ethical Concerns:**

["NO or VERY MINOR ethics concerns only"]

**Final Justification:**

As a reviewer, I believe the paper demonstrates a high level of quality in dataset construction, method design, and experimental results, and the authors have provided thorough and reasonable responses to the review comments. The authors have clearly addressed the issues and corrected formatting problems, further improving the quality of the paper. Given its technical novelty and depth of experiments, I recommend awarding a score of 5.

**Limitations:**

yes

**Paper Formatting Concerns:**

1.There is a typo in line 250.
2. The content is slightly crowded.

**Quality:**

3

**Strengths And Weaknesses:**

Strengths:
The proposed data construction scheme is reasonable, and the effectiveness of the proposed index screening can be seen from the supplementary materials. The open-source 210k Content-Style-Stylized datasets of the article is valuable to the community.

The designed end-to-end style migration scheme is quite intuitive, and the experimental results are very competitive and good.

The ablation experiments are reasonably designed, and the visualized experimental results are quite stunning.

This paper introduces the strengths and limitations of current methods, that is, there are some deficiencies in stylization of portrait scenes, providing directions for future exploration.


Weaknesses:
There are some details that may require further explanation.
The author's expressions are mostly clear and reasonable, but there are some minor formatting issues that need to be addressed.

---

> ### Author Rebuttal · Authors · 2025-07-31
>
> Response to Reviewer nay9
>
> We appreciate the reviewer's recognition and affirmation of various aspects of this study; your positive comments have provided significant motivation for us to further improve the research. Regarding the shortcomings you pointed out, we will attach great importance to them and handle them carefully.
> ***
> **Q1.** A total of 11,000 content images and 10,000 style images would theoretically yield 110 million potential combinations, yet the experimental output amounts to only 210,000 images. What explains this notable discrepancy between the theoretical combinatorial possibility and the actual production yield?
>
> **A1:** Firstly, the theoretical combinatorial calculation is a brute - force estimate based on the simple multiplication of the number of content and style images. In practice, not all combinations are viable or meaningful. Our experimental design incorporates a screening mechanism to eliminate such unpromising combinations. This screening process significantly reduces the number of combinations that are actually processed and output.
> Secondly, resource constraints play a major role. Generating and processing each image combination demands substantial computational resources, including GPU time, memory, and storage. Given the limited hardware resources available for our experiments, we are unable to exhaustively compute all possible combinations within a limited time. To optimize resource utilization, we prioritize the combinations that are more likely to produce high - quality and relevant results according to our predefined criteria.
>
>
>  ***
> **Q2.** Why injecting style features into ControlNet warrants a more detailed analysis.
>
> **A2:** We will elaborate on the reasons for choosing to pre-inject style features into ControlNet from two perspectives:
> (1) Style features require multi-level injection. Observations from InstantStyle experiments show that although style features can be condensed into certain two layers during downsampling, increasing the number of injection layers can significantly improve style similarity. Furthermore, ControlNet is essentially analogous to a part of the basic backbone network, such as the downsampling layers of U-Net. Moreover, for inpainting-oriented ControlNet, the downsampling layers themselves carry content information. Relying solely on feature injection in the upsampling layers may make it difficult to achieve high-quality style transfer. Therefore, injecting style features into the ControlNet branch, consistent with the injection method in InstantStyle, is intuitively reasonable.
> (2) The results of our ablation experiments are shown in the Table 1. The experimental results clearly demonstrate that injecting style features into the ControlNet branch can improve style similarity without causing style leakage or content loss.
>
> Table 1. Ablation studies style control.
>
>  | Metric          | w/o pre-inject style | feature |
>  |-----------------|----------------------|---------|
> | **CSD ↑**       | 0.5087               | 0.5146  |
> | **CAS ↓**       | 0.8462               | 0.8386  |
>
>  ***
> **Q3.** Table 2 requires a more detailed explanation. Table 2 references appear to be missing.There is a typo in line 250.
>
> **A3.** Table 2 presents human evaluation results comparing our method against StyleShot [12], InstantStyle [32], and StyleAligned. We aggregated evaluators' preference distributions across three categories: preference for our method ("better"), baseline ("worse"), or perceptual equivalence ("tie"). Key conclusions emerge: (1) Our approach demonstrates statistically significant superiority in style transfer quality, suggesting the efficacy of large-scale automated dataset construction for this domain; (2) Human preferences strongly correlate with quantitative metrics (CSD: Content Structure Distance, CAS: Content Awareness Score), validating their utility as perceptual consistency indicators.
>
> Thank you for the careful review. We will correct any typos.

---

> ### Author Response · Authors · 2025-08-05
> **Response**
>
> Dear Reviewer nay9,
>
> We have proactively addressed the concerns raised by the reviewers. We hope our responses will meet with your approval.
> We are keen to confirm whether the answers and experiments provided have sufficiently addressed the relevant questions and concerns. Please do not hesitate to raise any further inquiries you may have. Thank you sincerely for your dedicated efforts and ongoing support.
>
> With kind regards, The CSGO Author Team

---

> > ### Comment · Area_Chair_m7uf · 2025-08-05
> >
> > Dear Reviewer nay9,
> >
> > This is a gentle reminder to participate in the author discussion for your assigned paper(s). Engaging with authors is required before submitting the Mandatory Acknowledgement.
> >
> > The discussion deadline is August 8, 11:59 PM AoE. Please ensure you post at least one response in the discussion thread.
> >
> > Let me know if you encounter any issues.
> >
> > Best,
> > Area Chair, NeurIPS 2025

---

### Official Review · Reviewer_Exzg · 2025-07-01

**Clarity:** 2
**Significance:** 2
**Originality:** 3
**Rating:** 3
**Confidence:** 4

**Summary:**

This paper introduces IMAGStyle, a 210K diverse and precisely aligned triplet dataset for style transfer research. In addition, this paper proposes CSGO, a diffusion-based network that injects independent content and style features to support image-driven style transfer, text-driven stylized generation, and text-editing-driven stylized synthesis within a single architecture. Extensive experiments show that CSGO achieves state-of-the-art controllability and fidelity, demonstrating the critical role of structured synthetic data in unlocking robust and generalizable style transfer.

**Questions:**

1. Please provide a quantitative study showing how often the content LoRA and style LoRA truly isolate their intended factors.

2. Why not incorporate a perceptual style metric like CSD alongside CAS when choosing the target image?

3. AdaIN should divide by standard deviation, not variance.

**Ethical Concerns:**

["NO or VERY MINOR ethics concerns only"]

**Final Justification:**

1. They added quantitative decoupling study.
2. They added justification for CAS only screening articulated with alignment table.
3. Robustness of content and style separation is still uncertain at the instance level; averages do not guarantee absence of leakage.
4. Dataset construction still lacks a style sanity check.
5. Evaluation set and metrics are still not enough; no error bars or statistical tests; no perceptual or human study of CSGO.

The rebuttal does not fully resolve my concerns. Whether the proposed decoupling and dataset truly guarantee clean, scalable content-style separation. Therefore, I maintain my borderline reject rating.

**Limitations:**

Yes

**Quality:**

2

**Strengths And Weaknesses:**

Strengths:
1. They fill in the large-scale, high-quality data gap with explicit content-style-stylized supervision by IMAGStyle, a 210K triplet dataset.

2. They separate content and style conditioning into two independent injection branches, which jointly support image-driven style transfer, text-driven stylized generation, and text-editing-driven stylized synthesis.

3. They do plenty of ablation studies, including token counts, scaling factors, and branch designs. In addition, they compare with strong training-free baselines on both automatic metrics and human preference tests.

Weaknesses:
1. The pipeline assumes that each fine-tuned LoRA can be cleanly decomposed into content and style sub-LoRAs without quantitative evidence.

2. During dataset cleaning, the target image is chosen solely by the Content Alignment Score, but no style-alignment metric is applied. Therefore, triplets can retain wrong or mismatched styles.

3. AdaIN normalisation inside CAS divides by feature variance instead of standard deviation, making the score dimensionally inconsistent and casting doubt on every CAS-based claim.

4. The factorisation and filtering are not perfect, thus, style information may leak into the content branch.

---

> ### Author Rebuttal · Authors · 2025-07-31
>
> Response to Reviewer Exzg
>
> We thank reviewer for the constructive comments. We provide our feedbacks as follows.
> ***
> **Q1.** Please provide a quantitative study showing how often the content LoRA and style LoRA truly isolate their intended factors.
>
> **A1:**  Regarding the risk of B-LoRA.
> Our conclusion is first derived from the original literature [1], which verifies that B-LoRA has good decomposition properties. In addition, to address reviewer concerns regarding evaluation robustness, we implement the following protocol:
> (1) 300 content LoRAs and 300 style LoRAs are combinatorially paired, generating  stylized images per pair (total: 4500 images)
> (2) Three style consistency metrics are quantified:
>     (i) Mean style similarity between outputs and reference style images
>     (ii) Inter-group average of maximum style similarity
>     (iii) Inter-group average of minimum style similarity
> (3) Two established style assessment models provide metric validation:
>     - CSD [1]
>     - OneIG-StyleEncoder [2]
> (Both output normalized scores ∈ [-1,1], where higher values indicate stronger style fidelity)
>
> Table 1．The average style similarity of 300 groups of images, as well as the inter-group averages of the maximum and minimum similarity within each group.
> | Metric               | mean   | max    | min    |
> |----------------------|--------|--------|--------|
> | **CSD↑**             | 0.618  | 0.738  | 0.547  |
> | **OneIG-styleEncoder↑** | 0.679  | 0.815  | 0.611  |
>
> Table 2．The average CAS score of 300 groups of images, as well as the inter-group averages of the maximum and minimum similarity within each group.
> | Metric      | mean | max-mean | min-mean |
> |-------------|------|----------|----------|
> | **CAS ↓**   | 1.67 | 2.43     | 1.13     |
>
> As shown in Table 1, for unfiltered outputs, both assessors report mean style similarity ≈0.6, confirming that B-LoRA syntheses preserve high style fidelity regardless of content alignment quality.  On the other hand, we also present visualization results. The 10 sets of results shown in Figures 5-9 all indicate that the samples within these sets are relatively close to the original style images.
>
> Finally, we present in Table 2 the average CAS score among these 300 groups, the inter-group average of the maximum CAS score within each group, and the inter-group average of the minimum CAS score within each group. It can be observed that among the  generated results, we are able to obtain images with excellent alignment, so the inter-group average of the minimum CAS score within each group is 1.13. We show the visualization results of the CAS quantitative alignment in Figure 2 of the supplementary materials. It is quite consistent with the subjective perception that if the CAS is less than 1.2 points, the content can be considered relatively aligned. Therefore, from the experimental results of both aspects and the argumentation in the original article, it can be verified that B-LoRA can effectively decouple and combine content and style.
>
> [1] Somepalli, Gowthami, et al. "Measuring style similarity in diffusion models." arXiv preprint arXiv:2404.01292 (2024).
> [2] Chang, Jingjing, et al. "OneIG-Bench: Omni-dimensional Nuanced Evaluation for Image Generation." arXiv preprint arXiv:2506.07977 (2025).
>
>  ***
> **Q2.** Why not incorporate a perceptual style metric like CSD alongside CAS when choosing the target image?
>
> **A2:**  The reasons for not using the CSD score for screening are threefold:
> (1) B-LoRA can easily generate results that are close to the original style images. Empirical evidence from Table 1 confirms that there is high style fidelity between the synthesized images and the reference images (with an average similarity > 0.6). (2) Based on the intrinsic properties of B-LoRA decomposition: style features exhibit significantly higher decomposition stability, while content features show substantially greater variance. Therefore, it is more important to prioritize content accuracy when considering screening results. Visual verification results can be found in Figures 4 to 9 of the supplementary materials. (3)The screening results using CAS have the highest alignment with manual screening.
> To evaluate the alignment between different screening strategies and human selections, we conducted some verification experiments. We introduced a\*CAS + b\*(1-CSD) to represent the final score used for screening. We filtered the results using four strategies respectively and calculated the alignment with the manually screened results (top 1), as shown in Table 3. It can be observed that due to the instability in the content LoRA of B-LoRA, the results screened using only CAS are actually the closest to the manual screening results. Therefore, we choose to screen the dataset solely based on the CAS score.
>
> Table 3. The alignment between the screening results of different strategies and manual screening.
> | User   | a    | b    | Alignment Rate | User   | a    | b    | Alignment Rate | User   | a    | b    | Alignment Rate | User   | a    | b    | Alignment Rate |User   | a    | b    | Alignment Rate |
> |--------|------|------|----------------|--------|------|------|----------------|--------|------|------|----------------|--------|------|------|----------------|--------|------|------|----------------
> | **A**  | 1.0  | 0    | 85.3%          | **A**  | 0.75 | 0.25 | 65.3%          | **A**  | 0.5  | 0.5  | 53.3%          | **A**  | 0.25 | 0.75 | 16.7%          | **A**  | 0 | 1.0 | 9.7%          |
> | **B**  | 1.0    | 0  | 84.0%          | **B**  | 0.75 | 0.25 | 59.3%          | **B**  | 0.5  | 0.5  | 50.3%          | **B**  | 0.25 | 0.75  | 18.3%   |**B**  | 0 | 1.0 | 10.7%          |
> | **C**  | 1.0    | 0 | 82.7%          | **C**  | 0.75 | 0.25   | 58.3%          | **C**  | 0.5  | 0.5   | 47.7%          | **C**  | 0.25 | 0.75 | 15.3%  |**C**  | 0 | 1.0 | 13.0%          |
> | **D**  | 1.0    | 0  | 86.7%          | **D**  | 0.75 | 0.25  | 55.3%          | **D**  | 0.5  | 0.5  | 40.0%            | **D**  | 0.25 | 0.75  | 18.3%  |**D**  | 0 | 1.0 | 13.3%          |
>
>
>
>  ***
> **Q3.** AdaIN should divide by standard deviation, not variance.
>
> **A3：** We thank the reviewer for highlighting the precise terminology requirement; the manuscript has been corrected to explicitly reference "standard deviation" in the computation procedure.

---

> > ### Comment · Reviewer_Exzg · 2025-08-06
> >
> > Thanks for the rebuttal.
> >
> > 1. The rebuttal reports the average CSD computed over 300 × 300 LoRA pairs. However, a single mean can mask a long tail of cases where CSD is low and style leaks into the content branch. Even if only 10 % of the pairs are problematic, their gradients would still enter training while the mean would look healthy.
> >
> > 2. In each LoRA cluster, the target image T is selected solely by minimising CAS, a content-alignment score. Because no style metric is applied at this stage, T can end up visually indistinguishable from the content image C, offering little style supervision; The style exemplar S might itself exhibit a weak or mixed style, giving the network an ambiguous signal.
> >
> > Based on those concerns, I maintain the borderline reject rating.

---

> ### Author Response · Authors · 2025-08-05
> **Response**
>
> Dear Reviewer Exzg,
>
>  We would like to express our gratitude to the reviewers for their valuable feedback. We have carefully responded to the issues of concern raised by the reviewers:
> 1) Quantitative study of B-LoRA,
> 2)  why not incorporate a perceptual style metric like CSD alongside CAS when choosing the target image,
> 3)  and some typos
>
> We are keen to confirm whether the answers and experiments provided have sufficiently addressed the relevant questions and concerns. Please do not hesitate to raise any further inquiries you may have. Thank you sincerely for your dedicated efforts and ongoing support.
>
> With kind regards, The CSGO Author Team

---

> > ### Comment · Area_Chair_m7uf · 2025-08-05
> >
> > Dear Reviewer Exzg,
> >
> > This is a gentle reminder to participate in the author discussion for your assigned paper(s). Engaging with authors is required before submitting the Mandatory Acknowledgement.
> >
> > The discussion deadline is August 8, 11:59 PM AoE. Please ensure you post at least one response in the discussion thread.
> >
> > Let me know if you encounter any issues.
> >
> > Best,
> > Area Chair, NeurIPS 2025

---

### Official Review · Reviewer_QqDz · 2025-07-03

**Clarity:** 2
**Significance:** 2
**Originality:** 2
**Rating:** 4
**Confidence:** 4

**Summary:**

The paper addresses a key bottleneck in image style transfer research: the lack of large-scale, supervised datasets.  To solve this, the authors introduce a two-fold contribution. First, they design an automated pipeline to create and purify high-quality "content-style-stylized" image triplets. This pipeline is used to build IMAGStyle, a novel dataset comprising 210000 diverse image triplets, the first of its kind for style transfer research.

Second, leveraging this new dataset, the authors propose CSGO, an end-to-end trainable framework for style transfer. The CSGO model explicitly decouples content and style representations by injecting their features into different parts of a diffusion model. This single, unified framework is capable of performing image-driven style transfer, text-driven stylized generation, and stylized synthesis based on text editing, without needing to be retrained or fine-tuned for new styles. Experiments show that CSGO achieves state-of-the-art results in both style fidelity and content preservation compared to recent methods.

**Questions:**

Q1. CAS uses feature normalization to isolate content by removing style.  Could you provide further justification or ablation studies on the effectiveness of this approach compared to other potential methods for disentangling content and style?

Q2. Your ablation study shows that injecting style features into the ControlNet branch is beneficial.  Did you observe any trade-offs? Specifically, does this pre-injection of style into a content-focused module ever lead to undesirable content distortions or artifacts?

Q3. It has been mentioned that the model struggles with face stylization.  Could you elaborate on the specific failure modes? Is this primarily an issue with the underlying ControlNet, a lack of sufficient facial diversity in the IMAGStyle dataset, or the feature separation itself?

**Ethical Concerns:**

["NO or VERY MINOR ethics concerns only"]

**Final Justification:**

Authors have responsed my comments and provided additional results per request. I really appreciate it.

**Limitations:**

Please address the weakness points.

**Quality:**

2

**Strengths And Weaknesses:**

S1. The creation of the IMAGStyle dataset is a major contribution that directly addresses a long-standing challenge in the field, enabling new research avenues in end-to-end trainable style transfer models. The automated pipeline developed for its construction is also a valuable asset.

S2. The CSGO model is powerful due to its versatility. It handles three distinct and important stylization tasks (image-driven, text-driven, and text-editing-driven) within a single architecture, which is a significant advantage over methods that require separate models or fine-tuning for each style.

S3. The paper demonstrates through extensive qualitative and quantitative comparisons that CSGO outperforms a range of contemporary state-of-the-art methods. The model achieves a superior CSD score, indicating excellent style transfer, while maintaining strong content preservation.

W1. The data generation pipeline is built upon the B-LoRA method, which the authors themselves describe as "unstable".  Although a data-cleaning step using a so-called CAS score is implemented to filter out poor results, the final dataset's quality is fundamentally dependent on this initial, imperfect generation process.

W2. The CAS score, while is crucial for cleaning the dataset, relies on instance normalization to separate style from content features.  The assumption that this mathematical operation can perfectly and reliably remove "style" across all image types is strong and may not always hold true, potentially impacting the quality of the final IMAGStyle dataset.

W3. The paper acknowledges that it does not report error bars or conduct statistical significance tests for its experimental results.  This makes it more difficult to assess the consistency and reliability of the reported performance gains over other methods.

---

> ### Author Rebuttal · Authors · 2025-07-31
>
> Response to Reviewer QqDz.
>
> We thank reviewer for the constructive comments. We provide our feedbacks as follows.
>  ***
> **Q1.**  CAS uses feature normalization to isolate content by removing style. Could you provide further justification or ablation studies on the effectiveness of this approach compared to other potential methods for disentangling content and style?
>
>  **A1:** We verified through preliminary experiments that the proposed CAS has a very high similarity to manual screening. Secondly, we set up a strict screening mechanism and verified the reliability of the screening results. It is hoped that these experiments can alleviate the reviewers' concerns about CAS.
> First, to assess whether CAS aligns with human screening results, we randomly curated 300 content-style pairs and generated stylized results using the B-LoRA combination method introduced in our paper. For each of these 300 groups, each group contain  images as candiate image, we applied three distinct screening strategies to select the top-1 and top-2 results: (1) CAS scores computed using the CLIP visual encoder, (2) CAS scores computed using the DINOV2 model, and (3) manual human selection. The alignment rates between these computational methods (CLIP-based CAS and DINOV2-based CAS) and human choices are reported in Tables 1 (The visualization results (10 groups) are shown in Figures 5 to 9 of the supplementary material.).
>
> Table 1. Alignment results between CLIP-based CAS, DINOV2-based CAS and users.
> | Metric              | User A         | User A         | User B         | User B         | User C         | User C         | User D         | User D         |
> |---------------------|----------------|----------------|----------------|----------------|----------------|----------------|----------------|----------------|
> |                     | CAS-CLIP       | CAS-DINOV2     | CAS-CLIP       | CAS-DINOV2     | CAS-CLIP       | CAS-DINOV2     | CAS-CLIP       | CAS-DINOV2     |
> | alignment rate(top1)| 68.0%          | 85.3%          | 62.6%          | 84.0%          | 66.0%          | 82.7%          | 61.3%          | 86.7%          |
> | alignment rate(top2)| 86.7%          | 93.7%          | 82.6%          | 90.7%          | 80.7%          | 89.3%          | 82.0%          | 88.7%          |
>
> **Experimental results demonstrate that DINOV2-computed CAS scores achieve an high alignment accuracy.** **Representative visual examples are provided in Figures 9–14.** We analyzed the underlying factors contributing to this high alignment performance. As illustrated in Figure 3 of the supplementary material, we present the generated samples with their corresponding CAS-DINOV2 scores relative to the original content image. These scores exhibit strong consistency (aligning with subjective human assessment).
>
> Second, the technical details of the dataset screened based on CAS scores: Selecting the sample with the lowest CAS score per group constitutes merely one candidate solution. In practice, we establish a global threshold by aggregating the 25th percentile of minimum CAS scores across all groups. Consequently, for each group, only those containing at least one sample below this threshold are retained; all others are discarded. Furthermore, we implement globally normalized scores as the evaluation metric to maintain cross-dataset consistency. This systematic approach significantly enhances the reliability of our proposed automated dataset construction framework.
>
> ***
> **Q2.** Your ablation study shows that injecting style features into the ControlNet branch is beneficial. Did you observe any trade-offs? Specifically, does this pre-injection of style into a content-focused module ever lead to undesirable content distortions or artifacts?
>
> **A2:** In our experiments, injecting style features into the Controlnet branch can further improve the quality of style transfer without generating artifacts.  Since the proposed CSGO framework is trained end-to-end, unlike some training-free methods, it can correct some artifacts and abnormal states during training.
> The advantages of style feature injection in ControlNet are twofold:
> (1) Ablation studies confirm the efficacy of style feature injection in ControlNet, contingent upon consistent application during both training and inference phases.
> (2)Our pre-injection design preserves content integrity without inducing distortions or artifacts. As empirically validated in Figure 11 and Figure 12, content and style hyperparameters exclusively mediate the preservation fidelity of respective attributes in output images.
> ***
> **Q3.** It has been mentioned that the model struggles with face stylization. Could you elaborate on the specific failure modes? Is this primarily an issue with the underlying ControlNet, a lack of sufficient facial diversity in the IMAGStyle dataset, or the feature separation itself?
>
> **A3:** The suboptimal face stylization performance stems from two primary factors:
> First, the core limitation resides in training data paucity. As noted in Section 3, content prompts for our large-scale dataset derive from open-source salient object datasets (MSRA10K and MSRA-B). These collections exhibit strong bias toward non-human subjects, with severely insufficient portrait samples—directly causing deficient facial stylization data in our training corpus.
> Second, inherent instabilities in the B-LoRA module compound this issue. During content-style decoupled LoRA training, B-LoRA fails to consistently preserve critical portrait content—precisely retaining facial feature details constitutes an intrinsically challenging task [1].
> Finally, we clarify that while our method exhibits elevated failure rates in portrait style transfer, it maintains generalization capabilities within this domain—as explicitly demonstrated by the successful cases in Row 2 of Figure 4.
>
> [1]Frenkel, Yarden, et al. "Implicit style-content separation using b-lora." European Conference on Computer Vision. Cham: Springer Nature Switzerland, 2024.
>
> ***
> **Q4.** The paper acknowledges that it does not report error bars or conduct statistical significance tests for its experimental results. This makes it more difficult to assess the consistency and reliability of the reported performance gains over other methods.
>
> **A4:** To address reviewer concerns regarding result stability, we compute the final metric as the mean of five independent evaluations with standard error bounds (Table 4). These results confirm that the CSGO model achieves both statistically significant improvements over baselines and consistent performance across trials.
>  Table 4.More comprehensive quantitative comparison results between the proposed method and the other methods.
>
> | Method                | StyTR² [8] | Style-Aligned [14] | StyleID [7] | InstantStyle [32] | StyleShot [12] | StyleShot-lineart [12] | CSGO             |
> |-----------------------|------------|--------------------|-------------|-------------------|----------------|------------------------|------------------|
> | **CSD (↑)**           | 0.2695     | 0.4274             | 0.0992      | 0.3175            | 0.4522         | 0.4037 ±0.0174               | 0.5215 ±0.0162  |
> | **CAS (↓)**           | 0.9699     | 1.3930             | 0.4873      | 1.3147            | 1.5105         | 1.0910 ±0.0343                 | 0.8413 ±0.0270  |

---

> ### Author Response · Authors · 2025-08-05
> **Response**
>
> Dear Reviewer QqDz,
>
>  We would like to express our gratitude to the reviewers for their valuable feedback. We have carefully responded to the issues of concern raised by the reviewers:
> 1) The effectiveness of  CAS,
> 2) explanations of CSGO's limitations,
> 3) explanations of style feature injection into ControlNet,
> 4) and error bars
>
> We are keen to confirm whether the answers and experiments provided have sufficiently addressed the relevant questions and concerns. Please do not hesitate to raise any further inquiries you may have. Thank you sincerely for your dedicated efforts and ongoing support.
>
> With kind regards, The CSGO Author Team

---

> > ### Comment · Area_Chair_m7uf · 2025-08-05
> >
> > Dear Reviewer QqDz,
> >
> > This is a gentle reminder to participate in the author discussion for your assigned paper(s). Engaging with authors is required before submitting the Mandatory Acknowledgement.
> >
> > The discussion deadline is August 8, 11:59 PM AoE. Please ensure you post at least one response in the discussion thread.
> >
> > Let me know if you encounter any issues.
> >
> > Best,
> > Area Chair, NeurIPS 2025

---

### Comment · Area_Chair_m7uf · 2025-08-02

Dear Authors and Reviewers,

Thank you to the authors for the detailed rebuttal.

Reviewers, please read the responses carefully and post your reply as soon as possible to allow for meaningful discussion. Ideally, all reviewers should respond so the authors know their feedback has been considered.

Best regards,
AC

---

### Decision · Program_Chairs · 2025-09-17

**Decision:**

Accept (poster)

**Comment:**

This paper makes a clear and timely contribution by introducing the large-scale IMAGStyle dataset and the unified CSGO framework for content–style disentanglement in text-to-image generation. Reviewers agree that the dataset fills a long-standing gap and that the model demonstrates versatility and strong performance across multiple stylization tasks. While there were questions during the discussion regarding the stability of B-LoRA and the reliance on CAS for dataset cleaning, the authors provided additional quantitative evidence, ablation studies, and clarifications that addressed many of these points. Final ratings settled at two accepts, two borderline accepts, and one borderline reject, reflecting an overall positive reception. Considering its clear contributions and likely impact on the community, I recommend acceptance.